# Thrombin Cleavage of Osteopontin and the Host Anti-Tumor Immune Response

**DOI:** 10.3390/cancers15133480

**Published:** 2023-07-03

**Authors:** Lawrence L. Leung, Timothy Myles, John Morser

**Affiliations:** 1Division of Hematology, Stanford University School of Medicine, Stanford, CA 94305, USA; lawrence.leung@stanford.edu (L.L.L.); taniwha.whiro@gmail.com (T.M.); 2Veterans Affairs Palo Alto Health Care System, 3801 Miranda Avenue, Palo Alto, CA 94304, USA

**Keywords:** cancer, host anti-tumor response, metastasis, osteopontin, thrombin, tumor

## Abstract

**Simple Summary:**

Osteopontin (OPN) is a protein produced by immune cells, as well as many other cell types including tumor cells, that is highly modified with sugars and phosphates. OPN binds to receptors present on many cell types including cancer and immune cells and causes those cells to change properties. OPN is increased in the blood of cancer patients. OPN can be cut by thrombin, the major enzyme responsible for blood clotting, into two fragments that have different properties from intact OPN. These fragments suppress the host’s immune response to cancer, allowing the cancer to grow faster and to metastasize.

**Abstract:**

Osteopontin (OPN) is a multi-functional protein that is involved in various cellular processes such as cell adhesion, migration, and signaling. There is a single conserved thrombin cleavage site in OPN that, when cleaved, yields two fragments with different properties from full-length OPN. In cancer, OPN has tumor-promoting activity and plays a role in tumor growth and metastasis. High levels of OPN expression in cancer cells and tumor tissue are found in various types of cancer, including breast, lung, prostate, ovarian, colorectal, and pancreatic cancer, and are associated with poor prognosis and decreased survival rates. OPN promotes tumor progression and invasion by stimulating cell proliferation and angiogenesis and also facilitates the metastasis of cancer cells to other parts of the body by promoting cell adhesion and migration. Furthermore, OPN contributes to immune evasion by inhibiting the activity of immune cells. Thrombin cleavage of OPN initiates OPN’s tumor-promoting activity, and thrombin cleavage fragments of OPN down-regulate the host immune anti-tumor response.

## 1. Introduction

Osteopontin (OPN) is a secreted glycoprotein which was first discovered as a gene involved with activated T-cell responses in the resistance to human scrub typhus, and named an early T-cell activation gene (Eta-1) [1]. It was also identified in separate studies in bone [2,3]. The name, osteopontin, was coined based upon the rat bone sialoprotein to reflect that it is formed in the bone environment (*osteo*) and can form a bridge (*pon*) between cells and the hydroxyapatite matrix [4]. Subsequently, intracellular forms were identified and it is expressed in a broad range of tissue and cell types such as hematopoietic, fibroblastic, and immune cells. The expressed levels of OPN can be increased in different cell types by sundry inducers such as by the stimulation of the immune system or by inflammation [5]. OPN is present in mammalian milk at high levels and is among the top ten most abundant proteins in milk [6].

In control animals and people without disease, no physiological function has been proven for OPN. Mice with a full deficiency of OPN (OPN-KO) are fertile and undergo normal embryogenesis and development, and do not possess any phenotype before the challenge [7]. Even in the first report on OPN-KO mice, however, wound healing in response to a skin injury was shown to be modified in the absence of OPN. Detailed studies of pups born to OPN-KO females compared to those born to wild-type (WT) females show that the presence of OPN in milk promotes the development of the brain, intestine, and immunity, and leads to improvements in cognition, immune function, and intestinal development in early life [8]. When OPN is lacking in milk in OPN-KO mouse pups nursed by OPN-KO mothers, there is a reduced expression of OPN ligands, CD44, and integrins αv and β3 in the jejenum, resulting in lower signaling in the Erk, PI3K/Akt, Wnt, and FAK pathways [9]. Taken together, these data have resulted in recombinant OPN being evaluated as an additive for human infant nutrition [10].

OPN is encoded by a single copy gene on human chromosome 4 (4q13), called the secreted phosphoprotein 1 (*SPP1*) gene (Figure 1). The SPP1 gene contains seven exons [11] which result in 26 splice variants found in the ENSEMBL database (Figure 2). Of those 26 OPN mRNAs, only 6 are protein coding and 2 of those give rise to identical full-length proteins (osteopontin-a) that, in humans, consist of 314 amino acids (OPN residue numbering used in this review is based on the full-length human osteopontin-a protein including the signal sequence, unless otherwise specified). The predicted molecular weight of the unmodified peptide chain is 36 kDa but the protein exists in a variety of forms ranging from 41 to 75 kDa depending on the degree of post-translational modification (see below).

Several transcription factor binding sites have been identified in the OPN *SPP1* promoter including Ets1, β-catenin, runx1, C/EBPα, and AML1, all of which have previously been shown to have tumorigenic roles [13,14,15]. Transcription of the *SPP1* (OPN) gene is also regulated by miR-181a both in glioblastoma as well as during aging [16,17]. In addition to the regulation of the OPN expression at the transcriptional level, OPN mRNA is regulated by the interaction of the 5′ UTR with elongation factor 1A1 (EF1A1) [18].

A large number of polymorphisms have been identified along the *SPP1* gene and some have been associated with cancer and autoimmune diseases such as rheumatoid arthritis, asthma, and inflammatory bowel disease [19,20,21]. In urolithiasis, the OPN polymorphisms might affect the regulation of renal calcification and stone formation [22,23]. There are more than 300 variant sequences of the human *SPP1* gene catalogued in the genome databases. Of those variants, about 10 correspond to short deletion and insertion polymorphisms (indels), and the remainder are single nucleotide polymorphisms or variants (SNVs). 

Some of the OPN splice variants can be retained intracellularly [24] and the intracellular forms, osteopontin-c and osteopontin 5, are translated from an initiation codon in an exon accessed by alternative splicing [25]. Intracellular OPN interacts with BCL-6, a transcription factor, regulating the differentiation of follicular helper T cells [26]. The intracellular forms of OPN can accumulate in the nucleus where it might serve as a prognostic marker for breast cancer [27]. Intracellular OPN stabilizes TRAF-3, a component of the TNFR signaling pathway, by preventing its degradation via ubiquitin, leading to enhanced protection from viral infection by the increased production of IFN-b [28]. Thus, the intracellular forms of OPN can modulate immune system responses but, here, we focus on the role of thrombin cleavage of the extracellular forms of OPN.

Genome-wide association studies (GWAS) have shown a correlation of particular polymorphisms in the promoter region of OPN (−443 and −616) with the metastasis and prognosis of both gastric cancer and hepatocellular carcinoma [29,30,31]. A weak association of OPN SNVs with nasopharyngeal carcinoma has been found [32]. These studies, however, were carried out by examining registries and samples from the Chinese population and may not be generalizable. The risk of glioma is also related to polymorphisms at −155/−156 insG and −443 within the OPN promoter and those have been proposed for targeted cancer gene therapy [33,34]. When these polymorphisms were tested in reporter gene constructs, they all affected transcription from the OPN promoter [35]. Other SNVs in the OPN promoter as well as elsewhere in the gene have associations with cancer prevalence and some of those SNVs also occur within predicted transcription binding sites [36,37].

The mammalian *SPP1* gene encodes seven exons that are transcribed into OPN mRNAs of which exon 1 is not translated, while exons 2–7 encode the protein-coding sequences (Figure 3). Exon 2 encodes the signal peptide (SP) plus the first two amino acids of the mature protein. Exon 3 encodes sequences that can be phosphorylated at the serine residues and exon 4 encodes a proline-rich region and includes two glutamine residues (Q50 and Q52) that are potential transglutamination sites [38,39]. Exon 5 encodes another protein phosphorylation site. Finally, exons 6 and 7 encode the remainder of the OPN protein that includes the integrin-binding sites and thrombin and some other protease cleavage sites. A splice variant, osteopontin-b, has been identified that lacks exon 5. The different splice variants including osteopontin-a, -b, and -c, are expressed differentially in cancers [40,41]. A meta-analysis based on data in published literature supplemented with data from the TSVdb concluded that in lung cancer, OPN-a, OPN-b, and OPN-c were increased in comparison to normal tissue, while in breast cancer, OPN-c was elevated [42].

Within the region encoded by exons 6 and 7 is a conserved Arg-Gly-Asp (RGD) sequence that interacts with αvb1, αvb3, and αvb5 integrins, allowing binding to various components of the extracellular matrix as well as some cell types. The adjacent Ser-Val-Val-Tyr-Gly-Leu-Arg (SVVYGLR) sequence domain interacts with α4b1, α9b1, and α4b7 integrins only when this sequence is exposed at the C-terminus of the N-terminal OPN fragment in response to thrombin cleavage. It is functionally inactive in intact full-length OPN (OPN-FL). Thrombin cleavage generates an N-terminal fragment that includes both of these integrin-binding sites and gives rise to a C-terminal fragment containing calcium- and heparin-binding sites, which interacts with some variants of the cluster of differentiation 44 (CD44), such as CD44v3, CD44v6, and CD44v7 [7,8] (Figure 3A and Figure 4). In addition to thrombin cleavage, OPN can also be cleaved by other proteases including matrix metalloproteinases (MMP)-3 and MMP-7, plasmin, and cathepsin D. The proteolytic cleavages can render these domains more readily accessible, thus enhancing OPN binding efficiency to the integrins. The OPN variants generated by these enzymatic cleavages may retain their activities or acquire or lose additional functions [9,10].

Human OPN has conserved a similar domain structure and other features such as the integrin-binding sites and the thrombin cleavage site as other mammalian OPNs. Mammalian OPNs have strong homology over most of the protein sequence with OPNs from reptiles and birds except for sequences close to the C-terminus which, while highly conserved within mammals, are completely distinct in reptiles and birds, suggesting that these C-terminal sequences have unknown but different functions in mammals compared to reptiles and birds [43]. In addition to these tetrapods, the OPN gene exists in fish, supporting the concept that it was involved in the early evolution of bony skeletons and teeth.

This review will cover the role of OPN in cancer and the consequences of its cleavage by thrombin. OPN’s function in other areas such as fibrosis and bone and teeth development will not be included, although it should be noted that some of these other processes in which OPN is involved are partially related to those in cancer; for example, it encompasses OPN’s effects on myeloid cells during osteogenesis.

## 2. OPN Protein Structure

No full 3D protein structure has been determined for OPN by either nuclear magnetic resonance (NMR) or X-ray crystallography, and the predictions, including those from AlphaGo, suggest that it exists in solution as a random polypeptide chain. Attempts to determine its structure by NMR using labeling with ^13^C and/or ^15^N showed that OPN belongs to the class of proteins described as intrinsically disordered proteins [44,45]. Long-range NMR assignments show that the N- and C-termini of OPN interact and that those interactions can modulate the binding of OPN to its partners via the RGD integrin-binding site [46].

Although OPN exists in solution as an intrinsically disordered protein, it is hypothesized that binding to one or more of its partners induces a defined tertiary structure in the locality of the interaction that serves as the binding surface. This hypothesis has been confirmed by experiments in which structures of OPN bound to hydroxyapatite have been determined by NMR [47,48]. Thus, OPN may regulate the crystallization of calcium phosphate when hydroxyapatite is being laid down in bone formation. The phosphorylation status of OPN affects the local flexibility of the chain, with hyperphosphorylation causing increased flexibility and allowing more formation of local compact structures [49].

The structure of a peptide derived from close to the N-terminus of OPN (^40^VATWLNPDPSQK^51^) bound to a Fab fragment of an antibody has been determined by X-ray crystallography [50]. Bound to the antibody, the peptide has assumed a structure consisting of two tandem b-turns. A different antibody, that binds to fragment 1–127 of mouse OPN which does not contain the RGD sequence, was found to have altered the binding to full-length OPN with mutations in the RGD sequence, again suggesting that there are long-range interactions within OPN [51]. These long-range interactions were confirmed by NMR, showing that OPN in solution might have weak interactions between its N- and C-termini, thereby limiting easy access to the integrin- and other binding sites [46].

## 3. OPN Post-Translational Modifications

OPN is a member of the family of small integrin-binding ligand N-linked glycoproteins (SIBLINGs), a subfamily of the secreted calcium-binding phosphoproteins [52]. The genes encoding members of the SIBLING family are clustered together on chromosome 4 (Figure 1) and are believed to have originated by gene duplications from a single ancestral gene with subsequent divergence [53]. Each member of the SIBLING protein family has a minimum of one “acidic, serine-, and aspartic-acid-rich motif’ (ASARM). In addition, all members of the SIBLING family possess multiple Ser-X-Glu/pSer sequences (where X is any amino acid) which can be phosphorylated. When these sequences are phosphorylated, the negative charges on the phosphate groups allow them to bind either free calcium or the calcium contained in hydroxyapatite crystals.

Mature OPN protein is usually secreted from the cell via the canonical N-terminal signal sequence although some splice variants can be retained intracellularly [24]. During that secretory process, the nascent polypeptide chain has post-translational modifications such as O-linked glycosylations as well as sulfations and phosphorylations, as well as having the signal peptide removed (Figure 4) [54,55]. Glutamination can occur at Gln^50^ and Gln^52^. OPN sulfation may occur only in the context of bone, and, in experiments using radioactive sulfate to trace OPN in rat bone precursor cultures, it was unclear if the sulfate(s) are present as glycosaminoglycans or sulfotyrosines [56]. Subsequently, in experiments using OPN isolated from rat bone and analyzed by mass spectrometry, the site of sulfation was identified as rat Tyr^150^, which is homologous to human Tyr^181^ [57]. There was no evidence of sulfation of the O-linked glycans by mass spectrometry. The sites of the other modifications have been mapped onto the OPN protein sequence, showing that there are at least 5 sites at which O-linked glycosylation occurs and more than 36 possible phosphorylation sites [55,58,59]. The occupancy of those sites can vary depending on the cell type in which OPN is being produced [60,61,62]. In addition, one modification such as O-linked glycosylation may affect others such as how many sites are phosphorylated [54]. There are two consensus sequences (Asn-X-Ser/Thr where X is any amino acid) for N-linked glycosylation at ^79^AsnGluSer^81^ and ^106^AsnAspSer^108^ but there is no evidence for the presence of N-linked glycosylation on either Asn in the reported mass spectrometry analysis. Both of those serines can be phosphorylated, which would prevent glycan modification.

The phosphorylation of S^162^ adjacent to the integrin-binding sites in the sequence ^159^RGDSVVYGLR^168^ is carried out by FAM20C, also present in the microsomes [63]. The levels of FAM20C can vary, and, in the presence of expressed Ras, the levels of FAM20C and OPN phosphorylation are reduced [64]. The importance of these various post-translational modifications for the different OPN functions has not been fully explored.

## 4. OPN Binding to Extracellular Matrix Components 

OPN has roles as both a binding partner for various extracellular matrix (ECM) molecules including proteins, glycans, and hydroxyapatite, as well as functioning as a cytokine. Those binding partners interact with OPN via several conserved domains that allow contact with variant CD44 (vCD44), with heparan sulfate, with hydroxyapatite, with ICOSL, and with various integrins including the RGD domain that binds to α_v_b_3_ integrins. The OPN domains responsible for binding to the different partners are located in different parts of the OPN protein structure and thus may allow interactions with several partners simultaneously [43].

ICOSL is a binding partner of ICOS, both of which are type 1 plasma membrane proteins, and when bound to each other, a signaling cascade is initiated in both cell partners. Full-length OPN as well as both the N-terminal and C-terminal fragments formed by thrombin cleavage can bind to ICOSL [65]. The sites in OPN that bound to ICOSL were characterized by the limited proteolysis of the bound complex, followed by separation on SDS-PAGE with the sequence of the separated bands determined by LC/MS/MS. This showed that OPN contains two independent binding sites for ICOSL, one in the N-terminal thrombin cleavage fragment and the other in the C-terminal thrombin cleavage fragment. OPN acts as an inhibitor of the responses induced when ICOS binds to ICOSL, thereby regulating cells that utilize that binding for initiating changes in phenotypes such as migration and tumor angiogenesis.

Extracellular OPN possesses cytokine- and chemokine-like properties [5,66], for example, resulting in the modulation of leukocyte functions such as adhesion, survival, and migration [67,68]. In addition, under some conditions, full-length OPN or its fragments can bind to α_4_b_1_ and α_9_b_1_ integrins (see below). The binding of OPN to α_4_b_1_ integrin was not affected by the post-translational modifications of the OPN polypeptide chain [69]. In contrast, in a separate study, binding to b_1_ and α_v_b_3_ integrins was affected by the status of O-linked glycosylation and the phosphorylation of OPN [54,58].

As a matricellular protein, OPN is both a component of the extracellular matrix and circulates with pleiotropic cytokine- and chemokine-like functions and is expressed by osteoblasts and osteocytes, fibroblasts, hematopoietic, and immune cells (neutrophils, T and B lymphocytes, dendritic cells, NK cells, monocytes, and activated macrophages) [66]. In inflammatory conditions such as cancer and infection, its expression is markedly up-regulated [5]. Just C-terminal to the RGD sequence is the conserved thrombin cleavage site (at Arg^168^) that, when cleaved, exposes a previously cryptic integrin-binding site for α_4_β_1_ and α_9_β_1_ integrins at the new C-terminus, SVVYGLR [70]. The cryptic integrin-binding site revealed by thrombin cleavage is itself further processed by basic carboxypeptidases, reducing its integrin-binding affinity [71]. Thus, the proteolytic environment in which OPN is located will affect OPN’s integrin-binding activity, which is up-regulated by thrombin and, subsequently, down-regulated by basic carboxypeptidases.

The engagement of OPN with its ICOSL, integrin, and CD44 receptors on cells can lead to the initiation of a signaling cascade. The signaling response depends on the context with differing signaling cascades occurring in different cells. Also, the responses can differ depending on whether the cells are attached or in suspension and if the OPN is soluble or is a component in the extracellular matrix even in the same cell type. Binding of OPN to its receptors initiates the cell-signaling cascades by phosphorylation and the activation of kinases to transmit the signal to the nucleus [72]. In the nucleus, stimulation of various transcription factors occurs, resulting in modifications to gene expression and causing alterations in the cell’s phenotype. In the tumor microenvironment, cell interactions with OPN results in increased metastasis, angiogenesis, and tumor growth. During the process of tumorigenesis, the tumor cells themselves acquire mutations in signaling pathways, dysregulating them to promote growth [73].

c-Src kinase activity is induced after OPN binds to integrins αvb3 and αvb3, and that activity is necessary for the assembly of the complex of the integrins with EGFR signaling [74,75]. When TGF-α binds to EGFR, several signaling pathways are triggered including Ras-MAPK, PI3K, protein kinase C (PKC), and phospholipase-g [74,76]. EGFR signaling plays an important role in many cancers with epithelial origins [77]. Upon binding to CD44, OPN induces PI3K activity which, in turn, activates Akt, a kinase that has a central role in the promotion of cell motility by OPN [78]. This pathway can also cause resistance to apoptosis, shown in a BA/F3 murine B-cell line [79]. Signaling by OPN can be regulated by the tumor suppressor gene, PTEN [80].

Nuclear-factor-inducing kinase (NIK) is a member of the MAP3K that interacts with IKKα and IKKb, stimulating their activity [81]. OPN stimulates NIK-dependent NF-kB transcription through both IKKα/IKKb- and MAPK-mediated pathways [82]. There is an increased expression of NF-kB’s target genes including urokinase plasminogen activator (uPA), that is then secreted and which, in turn, activates MMP-9, leading to the degradation of the extracellular matrix which is a characteristic of cancer progression. OPN can also induce uPA expression via the c-Src transactivation of the EGFR, causing the activation of the transcription factor, AP-1 [76].

## 5. OPN Modulates the Innate and Adaptive Arms of the Immune System

When OPN interacts with cells via one or more of its receptors, signaling pathways are triggered that lead to changes in the cell phenotype. OPN and its receptors, the integrins, ICOSL and CD44, are widely expressed, suggesting that OPN plays a variety of roles in both normal physiology as well as pathophysiology. Many cells in the immune system both express OPN and respond to it, including T cells, B cells, NK cells, dendritic cells, and macrophages [83].

On T helper (Th) cells, OPN induces IL-17 by triggering integrin αvb3 and inhibits the secretion of IL-10 by binding to CD44 [84]. When OPN interacts with CD44 in Th cells, OPN induces the hypomethylation of IFN-𝛾 and IL-17a genes, enhancing the differentiation of Th1 and Th17 cells. In contrast, CD44 deficiency promotes hypermethylation of IFN-𝛾 and IL-17a and hypomethylation of the IL-4 gene, leading to Th2 cell differentiation [85].

As well as these effects on T cells, OPN modulates interactions between the innate and adaptive immune systems. When OPN triggers macrophages, IL-12 production is increased, leading to more Th1 differentiation [86]. In OPN-KO mice, there were fewer T-bet CD8^+^ T cells due to lower levels of IL-12, resulting in increased numbers of persistent long-term memory CD8^+^ T cells [87]. In plasmacytoid dendritic cells, OPN induces IFN-α which, in turn, is involved in the priming and polarization of CD4^+^ T cells [88]. 

Studies with a mouse strain deficient in the transcription factor IRF8 showed that the expression of OPN from the *SPP1* gene was repressed in CD11b^+^Ly6C^lo^Ly6G^+^ myeloid cells in the absence of IRF8 [89]. OPN acting via CD44 is a powerful repressor of T cells, functioning as an immune checkpoint. Colon epithelial cells express IRF8, with its levels being lower in colon carcinoma cells. Consequently, OPN production was increased in the carcinoma cells compared to the normal colon epithelial cells. This study suggests that OPN is functioning as a checkpoint in the development of host anti-cancer immunity [90]. 

Not only can OPN regulate the phenotype of immune cells, other cells present in tumors can modulate the production of OPN in a bidirectional feedback loop. Dendritic cells (CD1c^+^) produce OPN that suppresses T cells with the level of production of OPN by the dendritic cells depending on the status of mesenchymal stromal cells [91]. When the mesenchymal stromal cells are quiescent, the conditioned medium promotes OPN expression but when they are activated by treatment with inflammatory cytokines, OPN expression is inhibited by prostaglandin E_2_. In addition, OPN produced by dendritic cells can regulate their activation state, with the presence of OPN leading to larger cells that possess the increased production of costimulatory molecules and major histocompatibility complex class II antigens [92].

## 6. The Role of OPN in Cancer

There are several strong lines of evidence that OPN is involved with both primary tumor formation as well as metastasis. Those include the correlation between levels of OPN protein either in blood or in tumor biopsies with worse outcomes; data from single-cell RNA analysis of tumor biopsy samples detail the production of OPN by tumor cells as well as tumor-associated leukocytes, experiments exploring mechanisms upstream from the transcription of the OPN gene (*SPP1*) and downstream processes resulting from increased levels of OPN, as well as animal studies investigating the role of OPN in various models of cancer and, in particular, in genetically modified mice. OPN production by the primary tumor and its circulation may cause potential sites for metastasis to be preconditioned especially in the bone [93,94]. Taken together, the overall evidence is consistent with a key role for OPN in tumorigenesis and metastasis. A proof that OPN plays an essential function in these processes, however, is still lacking for the human disease. Most of these studies have not distinguished between different forms of OPN and, in general, have not investigated the role of thrombin or other protease cleavages of OPN in cancer (see below).

## 7. Increased Expression of OPN and Cancer

High serum or plasma levels of OPN have been associated with many human diseases, especially those that involve inflammation. These diseases include cancer in which high levels of OPN are prognostic for worse outcomes in several tumors, and its levels have been suggested as a marker for response to therapy [95]. Metastasis in melanomas, glioblastomas, and ovarian, lung, and breast carcinomas correlates with increased OPN levels [96,97,98]. The clinical utility of OPN as a marker in cancer therapy has been recognized by its inclusion in test panels for screening for eight tumors via liquid biopsy [99]. Indeed, when the expression of OPN was investigated in tumor tissues, the OPN gene *SPP1* was in the top 5% of overexpressed genes by microarray [100]. In several cancers, such as cervical squamous cell carcinoma and endocervical adenocarcinoma, colon adenocarcinoma, brain lower-grade glioma, liver hepatocellular carcinoma, lung adenocarcinoma, and rectum adenocarcinoma, the levels of OPN mRNA levels in tumor biopsies from patients predicted survival (Figure 5) [101,102].

In lung cancer, high blood OPN levels are a prognostic marker for worse outcomes such that patients with high OPN levels had a median survival of 57 months, while those with low OPN levels had a median survival of 102 months (hazard ratio = 1.57 (1.36–1.81), *p* = 5.390 × 10^−10^) [97]. OPN is overexpressed in lung cancer tissues and the tissue OPN is also associated with a poor prognosis and decreased survival rates in lung cancer patients. Serum OPN levels can be used as a prognostic marker in small-cell lung cancer and, in combination with CEA levels, as a diagnostic marker in non-small-cell lung cancer [103,104]. Overexpression of OPN is associated with poor outcomes in Alk1 fusion lung cancer patients who do not receive targeted therapy [105]. Plasma OPN levels serve as a better diagnostic marker than mesothelin levels for the diagnosis of mesothelioma [106]. In patients with lung cancer, circulating levels of OPN correlate with those of VEGF and MMP-9, suggesting that high OPN correlates with increased angiogenesis and, hence, the increased aggressiveness of the tumor [107].

After surgery in hepatocellular carcinoma patients, high plasma OPN levels correlated with a lower overall survival when compared to patients with low OPN levels (*p* = 0.001) [108]. In a meta-analysis of hepatocellular carcinoma that included 12 studies with 2117 patients published up to 2017, high blood OPN levels were associated with poor overall survival (OS; OR = 1.84; 95% confidence interval [CI] = 1.54–2.20; *p* = 0.0001) and disease-free survival (DFS; odds ratio (OR) = 1.67; 95% CI = 1.40–1.98; *p* = 0.0001) [109]. Furthermore, in the subgroup analysis, high levels of OPN in tissues assessed by immunohistochemistry detection and serum-based OPN by ELISA detection were both associated with worse OS (tissue: OR = 1.88; 95% CI = 1.53–2.31; *p* < 0.0001; serum: OR 2.38; 95% CI 1.58–3.59; *p* < 0.0001). In the same study, OPN expression was positively associated with the stage (OR = 5.68; 95% CI 3.443–7.758) and tumor size (size ≤ 5 cm vs. >5 cm; OR = 2.001; 95% CI = 1.036–3.867). Subsequent studies have confirmed that HCC patients could be stratified based on their OPN levels [110,111]. Extracellular vesicles released from a tumorigenic HCC cell line, HepG2, contained more OPN and less miR-181a, an miRNA that down-regulates OPN transcription, than extracellular vesicles from a non-tumorigenic line, WRL68 [112]. The current evidence indicates that OPN could serve as a prognostic biomarker and a potential therapeutic target for HCC.

High levels of OPN expression are associated with advanced stages of breast cancer, with lymph node metastasis, and with poor prognosis. Levels of OPN expression have been proposed as a predictive biomarker in the anti-EGFR therapy of triple negative breast cancer, but this proposal was based on in vitro experiments on breast cancer cell lines [113]. Higher OPN protein overexpression is found in breast tumors than in normal breast tissue, and higher plasma levels of OPN are positively associated with increased tumor burden and the shorter survival of patients [114,115]. In a meta-analysis of 10 breast cancer clinical studies, including a total of 1567 participants, both high levels of serum and tissue OPN indicated a poor breast cancer outcome. Moreover, OPN expression levels correlate with the expression of several other biomarkers, such as p53, other inflammatory markers, HER2, estrogen receptor (ER), and progesterone receptor (PR) [116,117]. In a study on tumor recurrence in tamoxifen-treated breast cancer patients, biopsies were evaluated for OPN mRNA expression by qPCR and OPN protein by immunohistochemistry [118]. OPN mRNA expression increased the risk of recurrence with an OR = 2.50 (95% CI; 1.30–4.82) for which, when adjusted for tumor grade, the HER 2 status and other treatments had an OR = 3.62 (95% CI; 1.45–9.07). OPN protein expression, however, was not associated with the risk of recurrence or with OPN mRNA expression, suggesting that OPN mRNA is a stronger prognostic marker than tumor tissue OPN protein.

A meta-analysis of ovarian carcinoma patients showed that they had higher levels of serum OPN than in healthy controls [119]. In this study, a subgroup analysis by ethnicity suggested that high levels of serum OPN might be the main risk factor for ovarian neoplasms in Asians (*p* < 0.001), but not in Caucasians (*p* > 0.05). Increased OPN mRNA expression levels are also associated with poor prognosis and decreased survival in ovarian carcinoma patients [120]. In an Indian population from South Asia, when OPN was compared with carbohydrate antigen 125 (CA125) as a diagnostic marker for ovarian carcinoma, OPN had a higher specificity than CA125 in detecting ovarian cancer. OPN can also differentiate better between benign and malignant ovarian cancer than CA125, suggesting its use as a diagnostic marker [121].

OPN expression levels are significantly higher in cutaneous melanoma patients than in healthy individuals, and higher OPN levels are associated with poor prognosis in melanoma patients [122,123]. In uveal melanoma, a proteomics study suggested OPN levels as a potential biomarker [124]. The V600E BRAF mutation is common in melanomas and is the target for the specific chemotherapy of melanoma. Levels of OPN and MMP9 were correlated and higher in melanoma patients than in controls but, in those patients with the V600E BRAF mutation, the levels of both OPN and MMP9 were not different than in those patients without that mutation [125]. Resistance to a BRAF inhibitor, the vemurafenib analogue PLX4720, was induced in four paired primary/metastatic cell lines, and, in each case, OPN expression was reduced in a manner similar to that seen in non-small-cell lung cancer [126,127]. Follow-up studies on melanoma cells resistant to both a BRAF inhibitor (encorafenib) and an MEK inhibitor (binimetinib) found that OPN was one of several genes whose expression was unaltered during a drug “holiday” [128]. These data support a role for OPN in the pathogenesis of melanoma and its potential as a marker, possibly to track resistance to chemotherapy.

Glioblastoma multiforme (GBM) is a highly aggressive and lethal brain tumor with poor prognosis, and its treatment remains a challenge for oncologists. OPN is highly expressed in GBM and has been implicated in the pathogenesis and progression of this malignancy such that increased serum levels of OPN correlate with poor prognosis [129,130]. OPN is also secreted into the CSF where its thrombin cleavage fragments have been detected [131,132]. OPN was one of five genes that distinguish highly invasive incurable glioblastoma from the less aggressive lower-grade astrocytoma in a study using differential expression to identify highly up-regulated genes [133].

These studies are representative of many others that, together, demonstrate a strong correlation between OPN levels in the blood with poor outcomes and support the use of OPN levels as a diagnostic marker. It also offers the possibility of following therapeutic outcomes in several cancers, but further studies are needed to validate that concept.

In all of these studies correlating OPN levels with outcomes in cancer and other diseases, the assays used to measure OPN levels in blood or tumor biopsies were commercial ELISAs that neither distinguish between the different splice forms of intact OPN-FL nor between the different proteolytically cleaved fragments of OPN. Indeed, in many cases, the exact OPN forms being detected are unknown. Future studies would be enhanced if they employed ELISAs that had been fully characterized with respect to the different forms of OPN being detected.

The role of the thrombin cleavage of OPN has not been investigated extensively in human cancer and, to our knowledge, the only studies in which intact OPN-FL has been specifically distinguished from the thrombin cleavage fragments have been that by Yamaguchi et al. investigating GBM [132]. Thrombin-cleaved OPN (OPN-arginine or OPN-R) and thrombin followed by basic carboxypeptidase double-cleaved OPN (OPN-leucine or OPN-L) were detected at higher levels in cerebrospinal fluid from cancer patients compared to non-cancer patients, and, in tissue biopsies, glioblastoma had increased OPN-R and OPN-L compared to tissues from patients with epilepsy.

## 8. OPN mRNA Expression in Tumor Cells and Tumor-Associated Cells

Evidence that OPN mRNA could be up-regulated in cells in tumors was first described in a rat osteosarcoma cell line [134] and has been confirmed in other cell culture models as well as in animal and human tumor biopsies [135,136,137,138]. One report demonstrated the role of the Wnt/β-catenin pathway in increasing OPN expression in colon cancer and its consequences on reduced survival when OPN is high [101]. Other signaling pathways, including hedgehog/Gli, can stimulate OPN mRNA expression [139,140].

Bulk RNA sequencing has been used to compare tumors grown in mice with genetic changes, such as a study on the role of NFAT2c in prostate cancer pinpointing the increase in OPN expression [141]. These studies and subsequent follow-up reports showed that both tumor cells and tumor-associated cells that form the tumor microenvironment could up-regulate OPN mRNA expression, but the precise cell types involved in human cancer were not elucidated.

Single-cell RNA sequence analysis overcomes that limitation and has shown that tumor-associated macrophages (TAMs), in particular, can increase OPN expression, leading to changes in their phenotype as well as influencing the overall outcome of the disease, such as in colorectal carcinoma and pancreatic ductal adenocarcinoma [142,143]. Importantly, the data from studies in colorectal cancer and cervical cancer suggest that OPN-producing TAMs may interact with tumor-associated fibroblasts to form the tumor microenvironment [144]. Use of the single-cell RNAseq technique allowed the demonstration of increasing heterogeneity and worse prognosis in more advanced hepatocellular carcinoma and showed that OPN expression correlated with tumor cell evolution and the reprogramming of the tumor microenvironment [145]. In gall bladder carcinoma, TAMs expressing OPN were associated with poor prognosis in PD-1-treated patients [146].

Characterization of TAMs in GBM by an RNA microarray found that they could not be classified into the simple M1 vs. M2 nomenclature [147]. Instead, the TAMs lay on a continuum between the M0 and M2 types of macrophages. That classification was originally derived from a comparison of macrophage responses that differ between mouse strains [148]. In this scheme, classic M1 macrophages are defined by the expression of the transcription factor, STAT1, and lead to an anti-tumor response via the presentation of tumor antigens to adaptive immune cells [149]. In contrast, alternatively activated M2 macrophages express intracellular STAT3 and the scavenger receptors, CD163 and CD204, as well as the mannose receptor, CD206, leading to the secretion of cytokines that suppress the host anti-tumor immune response [150]. 

Datasets of single-cell RNAseq from patients with non-small-cell lung cancer (NSCLC) who were smokers were first analyzed to identify clusters of cells using the expression of known markers for cell types, such as tumor cells with EPCAM and leukocytes with PTRPC (CD45) [151]. Then, the leukocytes were separated into classes using a similar method before splitting the myeloid cells into clusters of different cell types. Two types of immune-suppressive macrophages were found, one characterized by the expression of CCL18 which reduces the production of inflammatory factors, and the other categorized by OPN (*SPP1*) expression which promotes angiogenesis and matrix remodeling. Differences between samples from males and females were noted, with samples from males having a higher expression of the matrix proteins, OPN and *FN1*, as well as the complement protein, *C1QC*. Analysis of these three genes from other cohorts suggested that the higher expression of *FN1*, *SPP1*, and *C1QC* in immune cells correlates with a poorer prognosis. These differences might explain the sex-related difference in the prognosis of NSCLC, that is mainly due to variations in the immune response [152].

Another study analyzed single-cell RNAseq generated from early-stage NSCLC patients, finding that several cell types comprising a cell module that was named “the lung cancer activation module (LCAM^hi^)” could be identified. It consisted of PDCD1^+^CXCL13^+^ activated T cells, IgG^+^ plasma cells, and SPP1^+^ macrophages [153]. These tumor-associated macrophages are derived from monocytes and are distinct from alveolar macrophages. Using immunohistochemistry tumor markers for LCAM^hi^ cells to evaluate biopsies of NSCLC, this cellular signature was shown to be independent of overall immune cell content but did correlate with the tumor mutational burden, cancer testis antigens, and P53 mutations. High baseline LCAM^hi^ scores correlated with an enhanced NSCLC response to immunotherapy.

Lymph node metastasis can occur early in lung adenocarcinoma (LUAD) [154]. Biopsies from 16 patients with LUAD, half with metastasis and the other half without, were studied by single-cell RNAseq [155]. M0 macrophages were increased in the metastatic group while the epithelial–mesenchymal transition (EMT) pathway was increased in patients with higher M0 infiltration levels. OPN (*SPP1*) is also a gene involved in EMT that correlates with macrophage infiltration and M2 polarization. OPN was shown to be a potential marker for early lymph node metastasis in early-stage LUAD.

In glioblastoma, single-cell RNA sequence analysis has shown that tumor-associated fibroblasts are producing OPN [156]. This signaling network between the tumor cells, TAMs, and tumor-associated fibroblasts is not just present in primary tumors but is also established by colorectal cancer metastases in the liver [142]. In colorectal carcinoma, combining spatial transcriptomics with a single-cell RNA sequence allowed the colorectal cancer cells to be localized to the invasion front where the expression of HLA-G caused the induction of TAMs. These TAMS then produced OPN, which modified the tumor microenvironment to one with anti-tumor immunity, as well as increasing the potential for the proliferation and invasion of the tumor cells [157]. This type of network in which OPN acts as part of a communications interaction system between different cell types could be a common feature of cancers.

## 9. OPN in Tumor Cell Culture Models: Its Expression and Its Effects

Secreted OPN, whether produced by the cell itself and acting in an autocrine fashion or produced by other cells as a paracrine factor, stimulates a variety of effects on cells via the various different cellular receptors for OPN. In the context of tumors, OPN produced by tumor cells will modulate the behavior of those cells as well as cells in the tumor microenvironment. Effects of OPN on the tumor cells include changes in adhesion, migration, apoptosis, and proliferation. Cells in the tumor microenvironment are regulated by OPN to create a more tumorigenic niche. These changes include the generation of tumor-associated fibroblasts from resident fibroblasts and mesenchymal stem cells while inducing an immunosuppressive environment by the modulation of TAMs and regulatory T cells [158]. OPN also increases angiogenesis, critical for tumor survival and growth.

Studies have shown that OPN promotes lung cancer cell proliferation, invasion, and metastasis through several signaling pathways, including the PI3K/Akt and MAPK/ERK pathways. Overexpression of OPN in prostate cancer PC3 cells as well as non-transformed rat vascular smooth muscle cells led to the phosphorylation of ERK1/2, JNK, and p38 MAPK, three members of the MAPK/ERK pathway [159,160]. In human chondrosarcoma cells, OPN increases the phosphorylation of MEK, a MAPK, and FAK, and leads to the overexpression of MMP9, promoting more migration [160].

In a breast cancer cell line (MDA-MB-231), the knock-down of OPN with lentivirus led to reduced levels of integrin α_v_b_3_ with the subsequent inhibition of migration and invasion, along with a decrease in signaling in the PI3K/Akt pathway and an increase in autophagy-related proteins (LC3 and Beclin 1) [161]. OPN overexpression in prostate cancer epithelial cells (PC3, DU145, and LNCaP) leads to the activation of Akt via PI3K and ILK, which then regulates the expression of b-catenin [162].

OPN also contributes to the development of the resistance to both classical chemotherapy and targeted therapies in lung cancer. The inhibition of OPN expression or activity sensitizes lung cancer cells to chemotherapy and enhances the efficacy of targeted therapies such as EGFR inhibitors [163,164]. In contrast, one study in lung adenocarcinoma reported that OPN could cause EGFR phosphorylation, leading to sensitization to EGFR inhibitors [165].

OPN has also been found to interact with other signaling pathways and molecules that are involved in GBM, such as the Akt/mTOR pathway, matrix metalloproteinases (MMPs), and integrins. For example, OPN can activate the Akt/mTOR pathway and induce the epithelial–mesenchymal transition (EMT), a process that promotes cancer cell invasion and metastasis [166]. OPN can also induce MMP expression and activity, which, in turn, can degrade the extracellular matrix and promote tumor cell invasion and angiogenesis. When OPN is bound to integrins, such as α_v_β_3_ and α_v_β_5_, it can promote malignant astrocytoma cell adhesion and migration [167].

The addition of exogenous OPN protein to cell cultures changes the cell behavior in many functional assays such as ones for adhesion, apoptosis, migration, and proliferation. These effects are similar to those caused by the expression of OPN. Treatment of human mammary epithelial cells with exogenous OPN caused an increased induction of uPA expression, accompanied by an increase in migration [168]. In four human colon cancer cell lines, adding OPN caused a reduction in intertypic cell adhesion, an increase in migration via CD44, and an increase in invasion into Matrigel when OPN was included as a chemoattractant [169]. When these cell lines were transfected with a plasmid-encoding OPN and grown as subcutaneous tumors in nude mice, there was an increase in VEGF production in the tumor and more metastasis.

When HCC cell lines were exposed to different splice variants of OPN, OPN-a, OPN-b, and OPN-c, a migratory line, Hep3B, increased its migration in response to OPN-a and OPN-b but not to OPN-c, while in a non-migratory line, SK-Hep1, OPN-a had no effect and OPN-c inhibited migration [170]. In HepG2 cells, exogenous NO stimulates transcription from the OPN promoter and NO is also produced in the tumor microenvironment both by hepatoma cells and macrophages [171,172,173].

Angiogenesis is enhanced by the presence of OPN, resulting in the increased production of angiogenic factors such as VEGF or PGE_2_ by tumor cells and the stromal cells in the tumor microenvironment, as well as having direct effects on endothelial cells. The role of OPN in inducing VEGF and, subsequently, angiogenesis was demonstrated in both breast and colon cancer cells by siRNA knock-down [174,175]. In prostate cancer cells, OPN and MMP9 expression result in enhanced VEGF production [176]. Hypoxia is a common feature of solid tumors, leading to the induction of a large number of different proteins commonly via HIF1α. In breast cancer, OPN mRNA is induced by hypoxia, but its induction was independent of HIF1α [175]. The increased OPN leads to a positive feedback loop in which ILK and Akt are activated, causing the production of VEGF.

Endothelial precursor cells respond to the overexpression of OPN by accelerating angiogenesis in vitro via increased proliferation, migration, and tube formation. This is mediated by OPN interacting with integrin α_v_b_3_ on endothelial precursors, thereby leading to the activation of PI3K, Akt, and eNOS that, downstream, produce higher levels of NO. In mature endothelial cells, OPN treatment activates both the PI3K/Akt and the ERK1/2 pathways, inducing angiogenesis [177].

## 10. Characterization of the Functions of OPN in Animal Models of Cancer

Much of the mechanistic data on the role of OPN in cancer has been obtained from studies on genetically modified mice. In particular, OPN-deficient mice (OPN-KO) mice have been employed to identify the effects of OPN. Young OPN-KO mice have no obvious phenotypic changes compared to WT mice unless they are challenged in some way. As they age, some changes become apparent such as a reduction in the number of retinal ganglia cells and astrocytes in the retina in the absence of OPN [178].

In one of the first publications on OPN-KO mice, it was shown that when tumors were induced by repeated applications of a mutagen, N-methyl-N’-nitro-N-nitrosoguanidine (MNNG), to induce squamous carcinoma, OPN-KO mice had more tumors and metastases than WT animals [179]. There were more metastases in the OPN-KO mice than in WT mice but they were smaller. Tumor cells were derived from the MNNG-treated OPN-KO mice and transfected with either a plasmid-encoding OPN or one transfected with an empty vector to produce equivalent cell lines, one of which produced OPN, but the other did not produce any detectable OPN. These paired cells were inoculated into nude mice, that have normal OPN expression, where tumors from the cells expressing OPN grew slower than those with the empty vector. Immunohistochemical analysis of the tumors showed that there were more macrophages present in the slow-growing OPN-expressing tumors but there was a similar density of blood vessels. These data suggested that OPN can play different, apparently contradictory, roles in tumorigenesis as, during the chemical induction of tumors, OPN deficiency increases tumorigenesis, but in the cells derived from those tumors, the tumor expression of OPN led to reduced growth. OPN expression in the transfected cells increased host anti-tumor macrophage activity but did not alter angiogenesis. 

Confirmation of OPN’s role in down-regulating metastasis came from experiments in which OPN production in C5P2 cells that had been knocked down with ribozymes when injected into nude mice produced fewer metastases [180]. These C5P2 cells with reduced OPN expression were more susceptible to the toxic effects of NO produced by inducible NOS (iNOS) expressed in RAW 264.7 cells (a macrophage-like cell line).

In OPN-KO mice, chemical tumorigenesis was increased. This is a contradictory effect to that observed in other experimental settings, where the loss of OPN reduced the growth and metastasis of implanted tumors [179]. The tumor-enhancing effects of OPN deficiency on chemical tumorigenesis were corroborated in a two-stage model in which the mice were treated with 7,12-dimethylbenz(a)anthracene (DMBA) applied to the dorsal skin, followed by a twice-weekly application of 12-O-tetradecanoylphorbol-13-acetate (TPA) [181].

In OPN-KO mice compared to WT mice, B16 melanoma growth is suppressed via macrophages [182,183]. In the WT mice, OPN modulates the tumor microenvironment by interacting with α_9_b_1_ integrin on TAMs, inducing the cyclooxygenase-2-dependent production of PGE_2_, leading to increased angiogenesis. In contrast, a report from a different group showed that B16 growth was not inhibited by OPN in the s.c. flank model [184].

A breast cancer cell line, 4T1, that does not produce ICOSL but does secrete OPN, was compared in a Balb/c mouse model to 4T1^ICSOL^ cells that had been transfected with an expression plasmid for ICOSL [65]. The resultant tumor volume was lower in mice carrying tumors from 4T1^ICSOL^ cells than in 4T1 cells, but the number of metastatic nodules in the lungs was similar.

OPN production was suppressed in GL261, a murine GBM cell line, by increasing the expression of miR-181a via a lentivirus vector [17]. The transfected cells were implanted into the brains of immune-competent mice where they grew slower with more apoptosis and less proliferation. Overall survival was increased.

In the transgenic adenocarcinoma of the mouse prostate (TRAMP) model of prostate cancer on either a WT or OPN-KO background, OPN inhibits tumor growth in both the spontaneous primary tumor model and in a model in which those cells were transplanted into recipient mice [184]. In that model, OPN appeared to be unnecessary for the adaptive immune response but essential for the infiltration of NK cells. 

In models of mouse and human ovarian carcinoma implanted in WT mice, OPN produced by cancer-associated mesothelial cells was key to the development of chemoresistance [185].

Taken together, the data from these different types of studies on OPN and human cancer show that high levels of OPN expression are strongly associated in a variety of tumor types with worse outcomes than low levels. As well as tumor cells producing OPN, TAMs can be induced to produce OPN, altering their phenotype as well as affecting other cells in the tumor microenvironment such as fibroblasts and endothelial cells. Experiments in mice demonstrate that OPN is a key component of critical mechanisms in tumorigenesis, tumor growth, and metastasis, as well as modulating the host anti-tumor immune response. Similar proof of the key mechanistic role of OPN in human cancer is still missing. Additionally, the understanding of the role of the thrombin and other protease processing of OPN has not been fully explored.

## 11. Protease Cleavages of OPN

In mature OPN, adjacent to the RGD site is a unique conserved thrombin cleavage site (Arg^168^–R^168^) (Figure 6) that, when cleaved, generates two fragments, the N-terminal fragment that has the new C-terminus of R^168^ (OPN-R) and the C-terminal fragment that has the new N-terminus, Ser^169^ (OPN-C terminal fragment, OPN-CTF). This cleavage reveals the α_4_b_1_ and α_9_b_1_ integrin-binding site at the new C-terminus (SVVYGLR^168^) of OPN-R that is functionally inactive in intact OPN [43]. Early studies demonstrated that the adhesive properties of OPN were modified by the thrombin cleavage and that those were the consequences of modified integrin binding [70,186]. All secreted splice variants of OPN contain the thrombin cleavage site and will produce a N-terminal fragment with the functional integrin-binding sites following thrombin cleavage [24]. Apart from the activities associated with the new termini of the N-terminal OPN-R and C-terminal OPN-CTF fragments, all of the other OPN activities possessed by OPN-FL are still present in the fragments, although the strength of their interactions may differ when comparing the fragments to OPN-FL.

The rate of thrombin cleavage at that site is significantly lower than at thrombin’s canonical cleavage sites in blood coagulation substrates such as fibrinogen [187]. The thrombin cleavage of OPN, however, is probably occurring primarily in the extravascular space rather than intravascularly where there are high concentrations of serpins and other protease inhibitors that will inactivate thrombin. These inhibitors act stoichiometrically and would suppress thrombin activity until large quantities of thrombin had been generated. In the absence of those inhibitors, low levels of thrombin would be sufficient to readily cleave OPN. OPN is indeed a bona fide substrate of thrombin as the structure of OPN possesses sequences that bind to the two anion-binding exosites in thrombin and that thrombin cleavage of OPN is dependent on OPN binding to those thrombin anion-binding exosites [187].

The cryptic α_4_b_1_ and α_9_b_1_ integrin-binding site revealed in OPN-R following thrombin cleavage of OPN-FL can be further processed by the basic carboxypeptidases, particularly those in plasma: carboxypeptidases B2 (CPB2) and N (CPN). The generation of OPN-L eliminates the binding site for integrins α_4_b_1_ and α_9_b_1_ [71].

In addition to thrombin, many other proteases including plasmin, cathepsin D, and several matrix metalloproteases (MMPs), as well as a disintegrin and metalloprotease with a thrombospondin type 1 motif, member 8 (ADAMTS8) can cleave OPN at many sites, generating fragments that may be present in biopsy samples (Figure 6) [188,189,190]. OPN contains several cleavage sites for different MMPs, with one key site being immediately N-terminal to the thrombin cleavage site at Gly^166^. Cleavage at that site inactivates the binding of the OPN fragment to α_4_b_1_ and α_9_b_1_ integrins. Several MMPs have been implicated in cleavages of OPN including MMP3 (stromelysin-1), MMP7 (matrilysin), and MMP9 (gelatinase) [191].

In the context of this review, it is important to note that only thrombin out of all of the proteases in the blood coagulation cascade has been reported to cleave OPN, whether in the extrinsic (VIIa), intrinsic (IXa, Xia, and XIIa), or common pathways (Xa), nor have any of the complement proteases been shown to have OPN as a substrate. In a study on bovine milk, plasmin was shown to cleave OPN at several sites with the preferred site being Lys^170^, but it was not an efficient enzyme for cleaving OPN at Arg^168^, consistent with plasmin’s preference for lysine at the cleavage site [192].

We have shown that Arg^168^ represents a bona fide thrombin cleavage site, with a catalytic efficiency (*k*_cat_/K_M_) of 1.14 × 10^5^ M^−1^s^−1^, and cleavage requires engaging both thrombin anion-binding exosites I and II [187]. The kinetics of thrombin cleavage of OPN is much faster than those of the other enzymes that generate specific cleavage products from OPN. The importance of the thrombin cleavage site is underlined by its conservation in the OPN sequence [43,132]. The rate of OPN cleavage with the other enzymes reported to proteolyse OPN is much slower than with thrombin and, in vitro, the length of time for the reactions in the reports is much longer than for thrombin (hours vs. minutes) and the substrate enzyme ratio is often close to stoichiometric (1:1 vs.100:1). This suggests that thrombin will be the primary physiological enzyme and that the other enzymes will use as substrates the thrombin cleavage products.

Thrombin-cleaved OPN-R has enhanced α_4_β_1_-dependent cell-binding activity, displaying a five-fold increase compared to intact OPN-FL (Figure 7). This increased binding of α_4_b_1_ and α_9_b_1_ integrins to OPN-R is abolished when OPN-R is cleaved by either CPN, a constitutively active plasma carboxypeptidase, or CPB2 (also known as thrombin-activatable fibrinolysis inhibitor, TAFI, or carboxypeptidase U, CPU), converting OPN-R to OPN-L [71]. This second cleavage removes the C-terminal Arg^168^, converting the C-terminal sequence of OPN-R from SVVYGLR^168^ to SVVYGL^167^, generating OPN-L [71]. Cell binding to OPN-L is reduced from that observed with OPN-R to that found with OPN-FL. Thus, the thrombin cleavage of OPN-FL to generate OPN-R, and its subsequent processing by the carboxypeptidases to form OPN-L, represents an activation of the new integrin-binding capability, followed by its inactivation.

Although the properties of OPN-R have been well-studied, OPN-CTF has been less well-characterized. OPN-CTF also has activities different from that of OPN-FL. OPN-CTF binds to cyclophilin C and that complex can then interact with CD147 [193]. In a mouse mammary cancer cell model, OPN-CTF, together with cyclophilin C, activated CD147 signaling via Akt, leading to an increase in MMP2 production. This caused an increase in migration and was the first demonstration of a functional role for OPN-CTF that differs from the functions of OPN-FL. Furthermore, we identified a previously unrecognized pro-chemotactic sequence for dendritic cells (DCs) in intact OPN that spans the thrombin cleavage site (RSKSKKFRR). This sequence is disrupted following thrombin cleavage, but the loss of the pro-chemotactic activity in intact OPN is compensated for by the released OPN-CTF, which acquires a new conformation-dependent chemotactic activity towards DC [194]. Activity of OPN-CTF was confirmed in a human macrophage cell line (U937) where OPN-CTF was not as pro-inflammatory as OPN-FL and OPN-R [195]. Thus, thrombin cleavage of OPN occurs in vivo, and proteolysis regulates the functions of OPN with both generated fragments, OPN-R and OPN-CTF, acquiring different properties than possessed by OPN-FL.

Although, as described above, OPN is sensitive to cleavage by some proteases, it is resistant to other proteases. Importantly, for its function in milk for nutrition and other roles in infants, it is resistant to digestion by pepsin and gastric extracts [196]. O-linked glycosylation around the RGD site plays a role in conferring this resistance. Milk contains, in addition to full-length OPN, various N-terminal fragments [189], all of which would be expected to traverse the stomach without degradation before reaching the duodenum where they can modulate mucosal immunity. The N-terminal OPN fragments can be transported across the epithelial membrane by transcytosis [197].

Using ELISAs developed in our laboratory specific for OPN-R and OPN-L, we showed that levels of OPN-R and OPN-L are substantially elevated in the joint fluid of patients with inflammatory arthritis [67]. We also showed OPN-R and OPN-L present in tumor tissues and plasma from both glioblastoma (GBM) and non-GBM gliomas [132]. These findings demonstrate that the thrombin cleavage fragments of OPN are generated in human disease. The presence of OPN fragments that could be generated by thrombin was also demonstrated in bovine milk by the mass spectrometry analysis of the cleavage fragments of purified milk OPN, but several other cleavage fragments were also present [192].

The experiments described here on protease cleavages of OPN were all carried out in solution. OPN is a matricellular protein and the susceptibility of OPN to cleavage when it is a component of the extracellular matrix may be different both in protease specificity and the speed of the reaction from soluble reactions. 

## 12. Properties of the Thrombin Cleavage Fragments of OPN

From the earliest reports describing OPN, or secreted phosphoprotein 1, proteolytic cleavage of OPN was described and a trypsin-like enzyme was identified as the candidate responsible for that cleavage [198,199,200]. Thrombin treatment of the medium from cultures of Madin–Darby canine kidney cells showed the bands at ~30,000 and 20,000 m wt. on SDS-PAGE, of which the lower m wt. band was present in the medium without the addition of the enzyme and had homology to the C-terminus of OPN [201]. These OPN fragments had different properties from the full-length protein, in particular, in revealing a cryptic integrin-binding site for α4b1 and α9b1 integrins at the C-terminus of the new N-terminal fragment [202,203]. In addition to the cryptic integrin-binding site, the RGDS may have increased binding to the αvb1, αvb3, and αvb5 integrins [71].

The alterations in the binding of OPN ligands following thrombin cleavage results in functional consequences in both cell biology and in vivo experiments. Clearly, the binding of neutrophils is increased when binding sites for integrins α4b1 and α9b1 are revealed [204] and the adhesion of T-cells is increased. Those changes can be reversed, in some cases, by the treatment of OPN-R with a basic carboxypeptidase to generate OPN-L [67,71] demonstrating a mechanism for increasing activity following thrombin cleavage of OPN-FL and its reversal on the removal of the C-terminal Arg by basic carboxypeptidases. In plasma, a key basic carboxypeptidase precursor, proCPB2, is itself activated by thrombin in the presence of thrombomodulin to CPB2 [205,206], suggesting that this can be a self-regulating homeostatic mechanism for the activation and subsequent inactivation of OPN.

OPN-R and OPN-CTF have contrasting effects on cytokine production by human CD4^+^ T cells that also differ from OPN-FL [207]. All forms of OPN increased the secretion of IFN-g; only OPN-FL and OPN-R, but not OPN-CTF, enhanced the production of IL-17A; and only OPN-FL and OPN-CTF, but not OPN-R, caused the decreased expression of IL-10. The migration of human peripheral blood lymphocytes was promoted by OPN-FL, increased further by OPN-R, and not stimulated by OPN-CTF, while the adhesion of human umbilical vein endothelial cells was increased by OPN-FL and more by OPN-CTF, but not by OPN-R. These experiments demonstrated that OPN-R and OPN-CTF possessed different properties [207].

In animal studies, thrombin cleavage of OPN has been shown to occur in several models such as cardiac fibrosis, the hematopoietic stem cell (HSC) niche, microglia, and liver fibrogenesis [166,208,209,210]. In the pressure-overload cardiac fibrosis model, OPN expression is increased by 700-fold and thrombin cleavage of OPN is up-regulated, as shown by Western blots of the left ventricular lysates. In vitro experiments on cardiac fibroblasts showed that OPN-R up-regulated collagen 1 expression, a marker of fibrosis. Syndecan-4, a transmembrane heparan sulfate proteoglycan also up-regulated in this model, binds to OPN and inhibits the thrombin cleavage of OPN, protecting the heart. As the model progresses over time, syndecan-4 is shed, allowing the generation of OPN-R and increasing fibrosis.

The HSC niche is needed for maintaining HSC quiescence, retention, propagation, and differentiation, and requires OPN to regulate HSC proliferation negatively [211,212]. Fragments of OPN and not OPN-FL are the most abundant species present in the bone marrow where OPN-R acts as a chemoattractant and HSCs bind to it via their expressed α4b1 and α9b1 integrins [68]. In the absence of OPN, the homing ability of transplanted murine HSC is altered, which then causes a redistribution of bone marrow cells to the circulation and spleen and increases the mobilization of HSC upon granulocyte colony-stimulating factor (G-CSF) treatment.

Cells in the HSC niche produce all of the blood coagulation factors needed for thrombin generation, suggesting that circulating coagulation factors are not required for the activation of the coagulation cascade within the bone marrow [210]. Mature megakaryocytes with greater than 8N polyploidy are the major source of clotting factor V (FV), factor X (FX), and prothrombin (PT), needed for the assembly of the prothrombinase complex.

In gliomas, OPN expression is up-regulated compared to non-transformed astrocytes, processed to OPN-R, and that glioma-derived OPN-R is key in the activation of microglia and regulating macrophages [166]. The knock-down of OPN expression by shRNA in a rat glioma model reduced the size of the tumors. Using OPN-deficient mice, OPN was shown to be critical for macrophage infiltration and its absence caused smaller tumors with higher effector T-cell function [213].

The OPN thrombin cleavage fragments were tested in a mouse model of multiple sclerosis, experimental autoimmune encephalitis (EAE), induced by treatment with a peptide from myelin oligodendrocyte glycoprotein amino acids 30–55; the mice have an acute paralysis (relapse) and subsequently recover (remission) [207]. If OPN is injected during remission, the mice suffer a relapse [214]. When OPN-FL, OPN-CTF, and a mixture of OPN-R and OPN-CTF were injected, the mice had a strong relapse, while OPN-R caused a weaker relapse, and with a mutant of OPN that was resistant to thrombin cleavage, OPN-FL_R153S/S154F_, there was no significant relapse. This experiment demonstrated that thrombin cleavage of OPN-FL was driving the events leading to the relapse.

Overall, the data on the thrombin cleavage fragments of OPN suggest that they are key to OPN’s physiological and pathophysiological roles.

## 13. Thrombin-Resistant OPN_R153A_ Knock-in (KI) Mouse

To study the physiological and pathophysiological roles of thrombin cleavage of OPN in vivo, we generated a mouse in which the OPN gene had the mouse homolog of human Arg^168^, Arg^153^ (R^153^) mutated to Ala (A^153^) [215]. The OPN_R153A_ (OPN-KI) mice are fertile, development is normal, and they have no observed phenotype before the challenge. Notably, despite the reports of the role of thrombin cleavage of OPN in the hematopoietic stem cell niche, no differences were found in the complete blood count (CBC) when OPN-KI mice were compared to WT or OPN-KO mice. A possible explanation for this result is that, in the absence of OPN-R, there would be a smaller pool of HSCs, but normal mature cells could be formed from increased stem cell proliferation due to the absence of OPN-R and, therefore, decreased stem cell quiescence [68,210].

Based on earlier reports showing that B16 melanoma growth was reduced in OPN-KO mice compared to WT mice, we compared the OPN-KI mice to WT and OPN-KO mice in the s.c. model [182]. There was robust tumor growth in WT mice and significant suppression of tumor growth in the OPN-KO mice, confirming the earlier report that OPN has tumor-promoting activity. Strikingly, the thrombin-resistant OPN-KI mice showed a similar suppression of tumor growth as the OPN-KO mice. Similarly, in the hematogenous pulmonary metastasis model [216,217], there was a marked decrease in pulmonary metastasis in the OPN-KO and OPN-KI mice compared to the WT mice, shown by the number of metastatic lung nodules or by quantifying the melanin content in the lung tissues. This is the first in vivo demonstration that thrombin cleavage of OPN has a major pathophysiological role in cancer, and this indicates that, despite the presence of the other OPN functionalities in OPN_R153A_, including the RGD site for binding to integrins α_v_b_1_, α_v_b_3_, and α_v_b_5_, the vCD44-binding domain, and the heparin- and hydroxyapatite-binding sites, resistance to thrombin cleavage in OPN_R153A_ alone renders it incapable of supporting B16 tumor growth and metastasis, to an equal extent as the total absence of OPN. These data show that the prevention of thrombin cleavage of OPN in the OPN-KI mouse phenocopies the complete absence of OPN in the OPN-KO mouse in the B16 melanoma models. In these two B16 models, the OPN is derived from the host OPN since B16 cells do not produce OPN.

There was a marked increase in macrophages in the B16 tumor by immunohistochemistry in OPN-KO and OPN-KI mice compared to the WT tumor, suggesting that tumor suppression is mediated by tumor-associated macrophages (TAMs) that are not CD206^+^. This is similar to an earlier report which showed that tumors in OPN-KO mice had more CD206^+^ TAMs than in WT mice [179]. To test this mechanistic hypothesis, macrophages were depleted in vivo by treatment with clodronate liposomes [218,219], resulting in an abolition of the tumor suppression phenotype in the OPN-KI mice.

NOG mice have severe impairment of both innate and adaptive immune responses, including functionally defective T, B, and NK cells, as well as cytokine production [220]. To further validate the role of the immune system, NOG mice carrying the OPN-KI and OPN-KO genes instead of the WT OPN gene were tested in the two B16 melanoma models. In both models, tumor growth was similar in all three genotypes, confirming that an intact immune system is essential for the tumor suppression.

We investigated if changes occurred in the macrophage phenotypes in TAMs from OPN-KI and OPN-KO mice compared to WT mice by flow cytometry and found that, in addition to an increase in the total number of TAMs in B16 tumors in OPN-KI and OPN-KO mice, there was a switch from M2 TAMs (or CD11b^+^ CD38^−^CD206^+^EGR-2^+^) to TAMs with a different activation phenotype (CD11b^+^CD11c^−^CD206^+^Ly6G^−^) in these tumors (Figure 8). Both the M2 TAMS and those TAMs with the new phenotype express CD206, which might lead to worse outcomes in human cancers, but the meta-analysis showed that the data were very variable [221]. In this study, single-cell RNAseq was not used to analyze the TAMs, which prevents a more detailed description of them. There were similar numbers of infiltrating neutrophils, B cells, and T cells in tumors from OPN-KI and OPN-KO mice compared to WT mice.

Thus, macrophages are an essential step down-stream from the thrombin cleavage of OPN, and the thrombin cleavage fragment(s) of OPN cause the suppression of the host anti-tumor immune response, which is associated with a decrease in the total number of TAMs, resulting in increased tumor growth and metastasis. The blockade of thrombin cleavage of OPN in the OPN_R153A_ KI mouse relieves this suppression of the host anti-tumor immune response, allowing the increased infiltration of TAMs into the tumor but the fraction of M1 and M2 TAMs is reduced while the fraction of the new phenotype TAMs (CD11b^+^CD11c^−^CD206^+^Ly6G^−^) is increased, resulting in the suppression of B16 melanoma growth and metastasis in this model.

To further prove that thrombin cleavage of OPN is the critical step leading to the suppression of the host anti-tumor immune response, WT mice were either fed normal chow or chow containing dabigatran, the orally active direct thrombin inhibitor, following a protocol that produced a stable systemic anticoagulation effect in vivo. Dabigatran administration significantly suppressed B16 tumor growth and pulmonary metastasis in WT mice, replicating the OPN-KI phenotype and confirming that thrombin cleavage of OPN is the critical initiation step in this process.

These data on the reduced cancer growth and metastasis in OPN-KI mice are the first proof that thrombin cleavage of OPN is critical in the expression of the tumor-promoting properties of OPN. The available data do not show which cleavage fragment, be it OPN-R and/or OPN-CTF, is responsible, or, indeed, if further proteolytic processing of these fragments is required for the suppression of the host anti-tumor immune response.

## 14. Therapies Targeting OPN in Cancer

Given the data above, the cleavage of OPN by thrombin is a target for cancer therapy. Several approaches are possible for defining the precise target for the treatment, including inhibiting thrombin cleavage of OPN with anticoagulants, blocking the activities of either OPN-R or OPN-CTF, and antagonizing specific binding domains within OPN.

Anticoagulation has been studied as a treatment for cancer, including trials on warfarin and heparin. There is also a significant risk of thromboembolic events in cancer patients, and treatment as well as secondary prevention and low-molecular-weight heparin was used for these purposes. In addition, some cancer patients had previously been diagnosed with atrial fibrillation and prescribed warfarin for stroke prevention. Because of the lack of specificity of these earlier agents and the known disadvantages of i.v. or s.c. dosing for heparin and low-molecular-weight heparin [222], respectively, and the problem of maintaining consistent anticoagulant effects with warfarin and its low therapeutic ratio, trials of these agents were inconclusive. Argatroban, a direct thrombin inhibitor with low specificity, was shown to reduce the growth, invasiveness, and metastasis of a human breast cancer cell line in mice [223]. In preclinical studies, hirudin, a parenteral direct thrombin inhibitor, prevented metastasis of B16 melanoma cells in mice [224]. The newer direct oral anticoagulants, such as the thrombin inhibitor dabigatran and the factor Xa inhibitors, rivaroxaban and apixaban, circumvent those problems but the data so far in cancer in patients who do not have therapeutic or prophylactic indications for treatment with anticoagulants are not definitive [225].

The survival benefit of direct oral anti-coagulants (DOACs) over warfarin or heparin in cancer patients is unproven in clinical trials so far, maybe because of the sample sizes in the trials being too low and not allowing sufficient time for follow-up [226]. Currently, there are ongoing population-based studies suggesting that warfarin may be associated with improved overall survival in cancers and may protect against certain cancers. The anti-tumor effects of warfarin could occur through both coagulation-pathway-dependent and -independent mechanisms mediated by the inhibition of the Gas6-AXL signaling pathway. However, the use of anticoagulants to treat cancer patients is limited by the risks of bleeding. It is well-established that cancer patients have a higher incidence of major bleeds (6.5–18%) than patients without cancer (2–3%) on anticoagulation [227]. This increased bleeding cannot be fully accounted for by supratherapeutic anticoagulation and, instead, is likely to be related to abnormal tumor vasculature characterized by its immaturity and “leakiness” [228,229]. In addition, anticoagulation may be contraindicated in patients with primary brain cancers, such as GBM, and cancer patients with intracerebral brain metastasis, because of the risk of catastrophic intracranial bleed [230,231].

Anti-OPN mABs have been identified that bind different parts of the OPN polypeptide sequence and have been tested in preclinical cancer models and in the clinic for other indications. An mAB directed against the novel C-terminal site of OPN-R that is formed after thrombin cleavage, administered to a mouse model of human adult T-cell leukemia, resulted in the inhibition of tumor growth as well as tumor invasion and metastasis [232]. This antibody had a larger effect on tumor growth than an anti-OPN mAB that binds to the RGD sequence, while using both antibodies together had an additive effect.

The humanization of an mAB that binds to the sequence ^212^NAPSD^216^ adjacent to the calcium-binding domain of OPN was characterized in vitro to block the cell adhesion, migration, invasion, and colony formation of MDA-MB-435S cells, a human breast cancer cell line [233]. When tested in a murine breast cancer xenograft model, this mAB reduced the growth of the primary tumor and inhibited spontaneous metastasis to the lung, liver, and brain.

An anti-OPN mAB was identified that binds to the ^162^SVVYGLRSKS^171^ sequence which is next to the RGD sequence and spans the thrombin cleavage site of OPN [234]. This mAB blocked OPN binding to integrin α_v_b_3,_ inhibited thrombin cleavage of OPN-FL, and reduced cell migration. Its efficacy was demonstrated in vivo using a model of NSCLC in which the mAB, whether as a single agent or in combination with carboplatin, significantly inhibited the growth of large metastatic tumors in the lung.

Four anti-OPN mABs were identified based on their ability to block the OPN inhibition of T-cell activation [102]. Two of those clones increased the efficacy of tumor-specific CTLs in killing colon tumor cells in vitro, while, in vivo, they could suppress the metastasis of CT26 colon tumor cells in mice.

Anti-OPN antibodies have been tested in the clinic but, so far, with little success. ASK8007 is a humanized mAB that was derived from a murine mAB, 2K1, and is directed against the new C-terminus of OPN-R, SVVYGLR [235]. ASK8007 is similar to 2K1 in inhibiting OPN binding to the integrins αvb1, αvb3, αvβ5, α4β1, and α9β1, that interact with RGD and SVVYGLR sequences, respectively. A version of 2K1, C2KI, that is a chimera between the mouse 2K1 Fab domains and human Fc domain, cross-reacts with monkey OPN and treats collagen-induced arthritis in a cynomolgus monkey with no notable toxicological findings [236]. In a double-blind randomized phase IIA trial, rheumatoid arthritis patients received either ASK8007 or a placebo [237]. It was safe and well-tolerated but there was no clinical response and no changes in synovial sublining macrophages, although ASK8007 could be detected in the synovial fluid.

Use of mABs targeting OPN may have pharmacokinetic problems, resulting in the need for unrealistically high doses of the therapeutic antibody being required to reach efficacy because of the speed of the turnover of OPN in blood [238]. This study used heavy atom metabolic labeling to determine the half-life of circulating OPN in the blood of three volunteers and showed that the predicted t $\frac{1}{2}$ is 11 min. The pharmacokinetic/pharmacodynamic relationship was used to calculate the required dosing of anti-OPN mABs based on assumptions that the mean blood levels of OPN would be 444 ng/mL with a range of 112–1740 ng/mL [239] and the mAB would be required to achieve a 90% reduction in circulating OPN. The study used four different assumptions about the mAB properties, one with standard mAB properties and the other three that had been engineered in different ways to improve the mAB pharmacokinetics (PK) and pharmacodynamics (PD). Although the calculated dosing regimens are theoretically possible with the most favorable assumptions, the authors’ conclusion was that the high amount of drug substance and inconvenience of administration would generally preclude the type of dosing regimen necessary.

## 15. Conclusions

Although there are extensive data on the biochemistry and cell biology of OPN, there is still no physiological function of OPN that explains its evolutionary conservation. Many defects have been described in OPN-KO mice but those are not ones that would cause selection pressure to retain the homologous sequence and its functional domains within mammalian genomes. 

There are many lines of evidence showing that OPN is very strongly associated with cancer. The data on the prognostic power of OPN levels in blood and in tumor biopsies in several indications are strong, and studies are being conducted now to confirm that OPN may also be a useful marker for following the response to treatment. 

Several cell types in the tumor microenvironment, in addition to the tumor cells themselves, can produce OPN. In that niche, OPN-producing cells prominently include TAMs whose phenotype is regulated by OPN, as well as critical cells for the establishment of a flourishing tumor niche such as tumor-associated fibroblasts and endothelial cells.

Engagement of OPN with its different receptors leads not only to mechanical consequences in the bone and probably elsewhere, but also to initiating signaling cascades within cells. These cascades ultimately lead to the induction of the transcription of specific genes that can alter the phenotype and behavior of the cells, as well affecting cells in its environs. Intriguingly, in some situations in tumors, the presence of OPN causes the increased production of OPN, creating a feed-forward loop.

A variety of enzymes can cause the proteolysis of OPN but, in vitro, the rates of cleavage are very often too slow and the reactions are often carried out using a high enzyme/substrate ratio. The enzyme can clearly cleave OPN but the conditions employed mean that the cleavage is unlikely to be physiologically relevant. OPN is a physiological substrate for both plasmin and thrombin and they can cleave it at a significant rate using a low enzyme/substrate ratio. The cleavage products of both plasmin and thrombin have been detected in various biopsy samples, showing that those cleavages can occur in vivo.

OPN is a pathophysiological substrate of thrombin based on the recent data employing the OPN-KI mice that showed, in a melanoma model, that the initiation of OPN’s tumor-promoting effects in vivo depends critically on thrombin’s cleavage of OPN [215]. Tumor suppression is similar in OPN-KI mice and OPN-KO mice and the direct thrombin inhibitor, dabigatran, replicates the reduction of tumor growth in WT mice, showing that intact OPN-FL does not, itself, promote tumor progression. This phenotype in the OPN-KI mouse is lost when the OPN_R153A_ (OPN-KI) mutation is on the immune-deficient NOG-background mice and is reversed following macrophage depletion by clodronate treatment of OPN-KI mice. These data indicate that the tumor suppression resulting from the thrombin cleavage of OPN is mediated by changes in TAMs. 

Rivaroxaban, a DOAC-targeting factor Xa in the clotting cascade, also promotes host anti-tumor immunity. In a murine model of fibrosarcoma using T241 cells, monocytes and F4/80^+^ TAMs expressed tissue factor, factor VII, and factor X, allowing the activation of factor X to factor Xa and probably generating thrombin downstream [240]. Treatment with rivaroxaban inhibits factor-Xa-protease-activated receptor 2 signaling in those TAMs and modulates the phenotype by reducing the fraction of MrcI^+^CD204^+^ TAMs. The population of T cells was also regulated upon rivaroxaban treatment, reducing the immune-suppressive regulatory CD4^+^ FoxP3^+^ T cells and increasing tumor-killing granzyme B^+^ CD8 T cells. This study is consistent with the inhibition of factor Xa preventing the formation of thrombin, thereby blocking the cleavage of OPN by thrombin and avoiding the reduction in host anti-tumor immunity.

Commonly, tumor cells express a variety of procoagulant molecules that initiate the coagulation cascade and lead to thrombin generation with cross-talk between coagulation activation, inflammation, and cancer cell biology [241,242,243]. This suggests that treatment with anticoagulants could be beneficial for cancer patients. In a model of metastasis of B16 cells, hirudin, a parenteral direct thrombin inhibitor, reduced metastasis to the lungs [224]. When the effects of hirudin were compared to low-molecular-weight heparin in models of metastasis using B16 and K1735 melanoma cells and CT26 colon carcinoma cells, differential effects were observed [244]. Growth of B16 cells was completely halted in mice treated with hirudin while there was no effect on the K1735 or CT26 development of lung metastasis, and low-molecular-weight heparin did not change outcomes in the K1735 model. Thus, the outcome of treatment of tumor models with anticoagulants may depend on the properties of the cancer cells being studied. 

In human cancer patients, a Cochrane systematic review showed that there is a survival benefit of parenteral heparin, particularly in those with limited small-cell lung cancers [245]. The conclusion that anticoagulation has a direct beneficial effect on cancer patient survival remains controversial [246]. So far, the analysis of outcomes for cancer patients treated with anticoagulants have utilized data from older drugs such as heparin, low-molecular-weight heparin, and warfarin, which have off-target effects and whose therapeutic ratio are not as good as the newer DOACs [247]. Dabigatran, rivaroxaban, and other DOACs could be used as adjuncts to conventional chemotherapy in cancer treatment. A higher incidence of bleeding complications, however, is observed in cancer patients when they are anticoagulated, possibly caused by the abnormal tumor vasculature [227].

Our finding that thrombin cleavage of OPN modulates macrophage subsets is relevant in other pathological conditions. One indication that has been shown to have a strong correlation with OPN is metabolic-associated fatty liver disease (MAFLD) and the subsequent non-alcoholic steatohepatitis (NASH). Blocking OPN with an mAB has a protective effect in a mouse NASH model [248]. In addition, thrombin-cleaved OPN has been reported to induce collagen production in cardiac fibroblasts and may play a role in cardiac fibrosis [249].

Our data support the notion that thrombin cleavage of OPN, be it derived from the tumor and/or the host, leads to the suppression of the host anti-tumor immune response, and cancer cells exploit this mechanism to enhance their survival.

## 16. Future Directions

A major problem in studies on associations between OPN levels and outcomes in various indications is that the ELISAs routinely used have not been fully characterized and validated against the variety of OPN fragments present in biopsy samples. For many of the commercial ELISAs, the antigen used to raise the antibodies and the location of the binding sites for the antibodies are trade secrets. The specificity of the antibodies and ELISAs has often only been shown using recombinant OPN or the antigen used to originally raise the antibody. 

Using a recombinant OPN, the dose–response curves of the commercial ELISAs show a range of sensitivities. The data, however, is often generated in buffer and not in a relevant matrix for biopsy samples. Importantly, the relative sensitivities in the ELISAs of different OPN fragments compared to OPN-FL are not available. With all these problems taken into consideration, researchers should be aware of the possibility of generating misleading results and should be accurate in their reports on the ambiguities caused by the currently available ELISAs. The manufacturers of the ELISAs have a responsibility to provide more information about the origin of the reagents and how much validation, if any, has been carried out on their ELISAs. In the longer term, the manufacturers should improve the quality of the ELISAs, especially if the field is to start the use of OPN levels as a prognostic marker. 

Although the evidence that thrombin cleavage of OPN is important for the host anti-tumor immune response, the source of the thrombin is unknown. Coagulation factors are present in tumors, but there need to be studies to identify which cells produce them. The tumor environment is described as procoagulant; it is not known which cells support the assembly of the Xase and prothombinase complexes required for efficient thrombin generation. 

The extensive data now available show that thrombin cleavage of OPN plays a key role in regulating the tumor microenvironment and, in particular, the host anti-tumor response via macrophages. This one step of the thrombin cleavage of intact full-length OPN begins a process which leads to fewer M1 and M2 TAMs infiltrating into the tumor and the suppression of the host anti-tumor immune response, thereby generating conditions favoring tumor growth and metastasis. Inhibiting this crucial step of thrombin cleavage or its downstream consequences would result in the enhancement of the host anti-tumor immune response and tumor suppression. This is a target for the discovery of novel therapeutic strategies for cancer and, potentially, other indications.

## Figures and Tables

**Figure 1 cancers-15-03480-f001:**
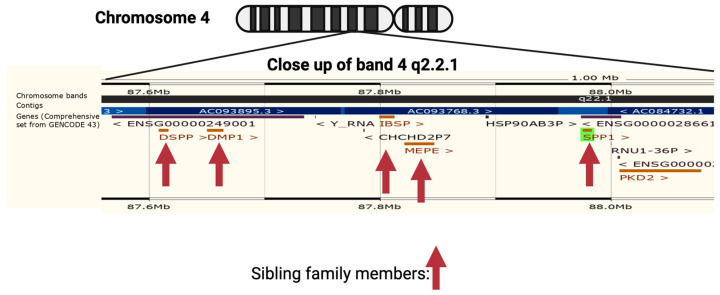
Neighborhood of osteopontin (*SPP1*) gene on human chromosome 4. The genes encoding members of the small integrin-binding ligands with N-linked glycosylation (SIBLING) family are marked with a vertical arrow: bone sialoprotein (*BSP*), dentin matrix protein 1 (*DMP1*), dentin sialophosphoprotein (*DSPP*), matrix extracellular phosphoprotein (*MEPE*), and Osteopontin (*Spp1*). Created with BioRender.com with data from ENSEMBL v109, human assembly GRCh38.p13 [12].

**Figure 2 cancers-15-03480-f002:**
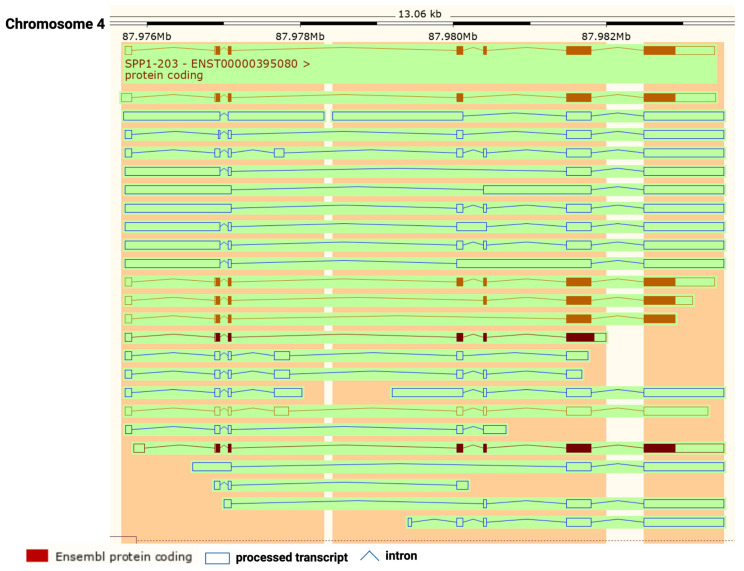
Osteopontin (OPN) splice variants. Transcripts from *SPP1* gene with protein-coding exons shown in filled-in brown boxes. Other processed transcripts that are not translated are shown as blue open boxes. Introns are shown as blue lines. Created with BioRender.com with data from ENSEMBL v109, human assembly GRCh38.p13 [12].

**Figure 3 cancers-15-03480-f003:**
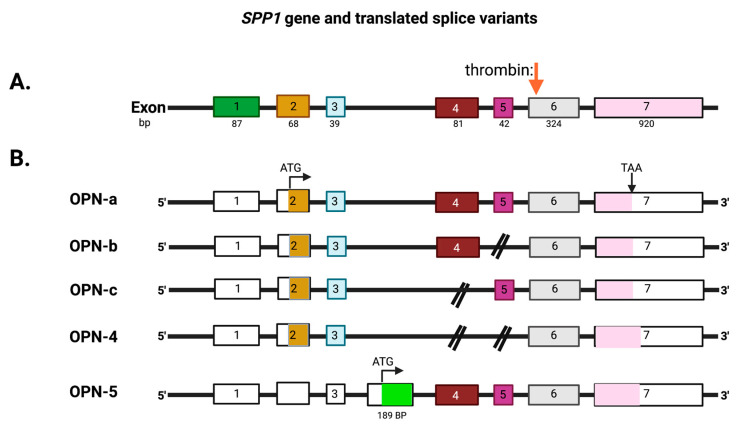
Exons in the osteopontin gene. (**A**) OPN (*SPP1*) exons on chromosome 4. (**B**) The exons that are translated from the OPN mRNAs into the extracellular forms, OPN-a, OPN-b, OPN-c, and OPN-4, and the intracellular form, OPN-5, with its unique exon (in green). The full-length protein of 314 amino acids, OPN-a, is translated from the upper splice variant. Thrombin cleavage site: red arrow. ATG: start codon. TAA: stop codon. Created with BioRender.com.

**Figure 4 cancers-15-03480-f004:**
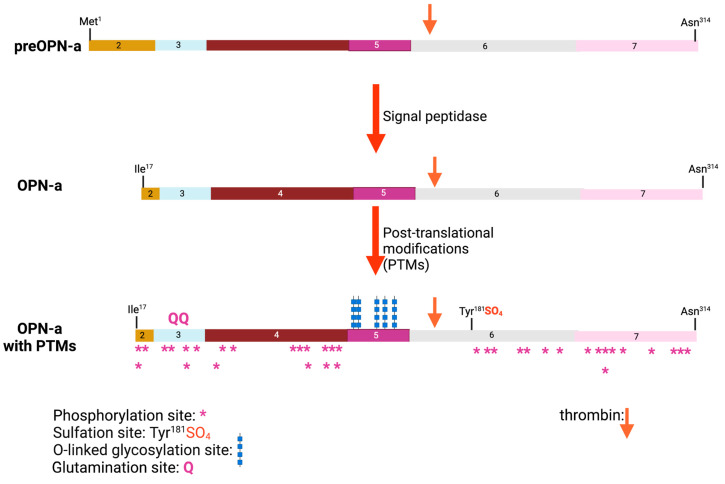
Osteopontin post-translational modifications (PTMs). PTMs are marked for glutamination, O-linked glycosylation, phosphorylation, and sulfation, and the thrombin cleavage site is marked with a red arrow. The numbers represent the exon encoding that sequence. Created with BioRender.com.

**Figure 5 cancers-15-03480-f005:**
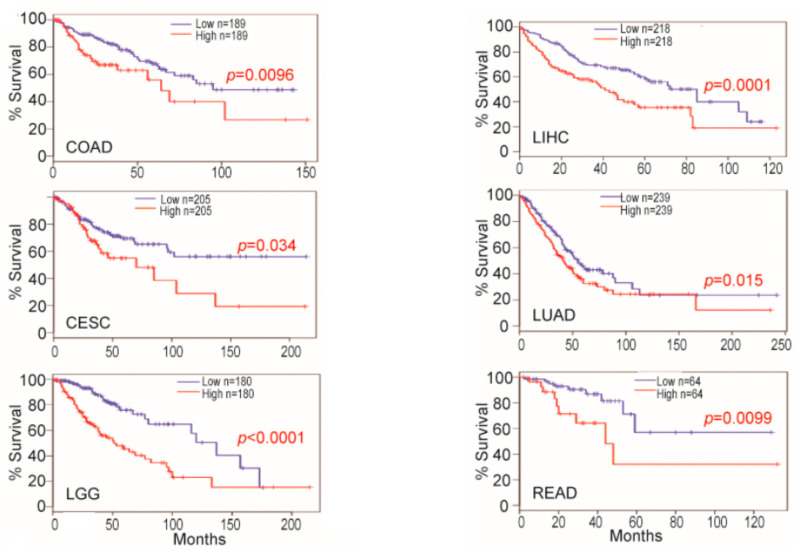
Osteopontin (OPN) mRNA levels in non-neoplastic colon and colon carcinoma predict survival. OPN mRNA expression level and survival datasets were extracted and plotted for survival. For each indication, the values considered high and low levels of OPN are shown in ng/mL. Blue line: survival of low OPN group, red line: survival of high OPN group. CESC: cervical squamous cell carcinoma and endocervical adenocarcinoma; COAD: colon adenocarcinoma; LGG: brain lower-grade glioma; LIHC: liver hepatocellular carcinoma; LUAD: lung adenocarcinoma; READ: rectum adenocarcinoma. Data from [102]. Created with BioRender.com.

**Figure 6 cancers-15-03480-f006:**
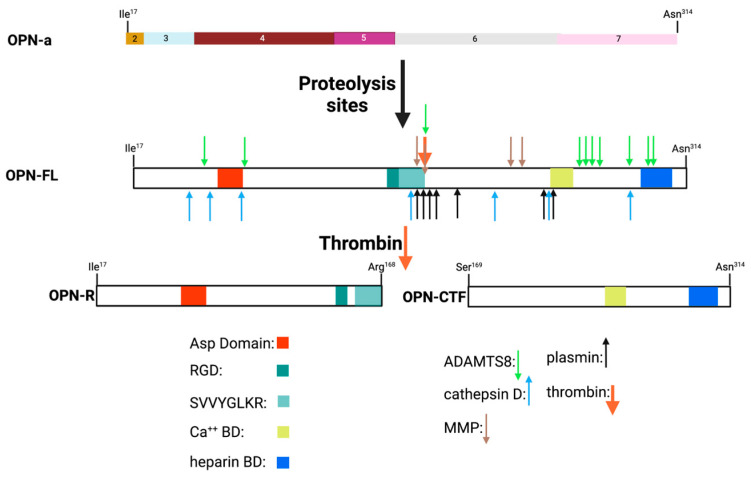
Proteolytic cleavages in osteopontin and the thrombin cleavage fragments. Vertical arrows mark site of enzyme cleavage. Asp D: aspartic-acid-rich domain, Ca^++^ BD: calcium-binding domain, heparin BD: heparin-binding domain, RGD: ArgGlyAsp integrin-binding site, SVVYGLR: cryptic integrin-binding site. OPN-FL: full-length OPN, OPN-R: N-terminal fragment generated by thrombin cleavage with C-terminal Arg^168^, OPN-CTF: C-terminal fragment generated by thrombin cleavage with N-terminal Ser^169^. Created with BioRender.com.

**Figure 7 cancers-15-03480-f007:**
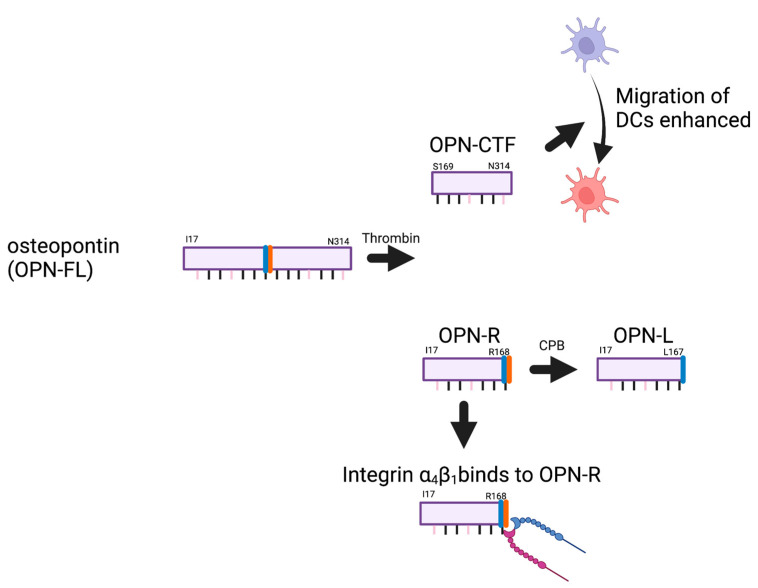
Thrombin cleavage of osteopontin. Thrombin cleaves full-length osteopontin (OPN-FL) into OPN-R, the N-terminal fragment, and OPN-CTF, the C-terminal fragment. OPN-CTF enhances chemotaxis of dendritic cells (DCs). OPN-R possesses a new α_4_b_1_ (and α_9_b_1_) integrin-binding site that is destroyed upon basic carboxypeptidase (CPB) cleavage to OPN-L. Created with BioRender.com.

**Figure 8 cancers-15-03480-f008:**
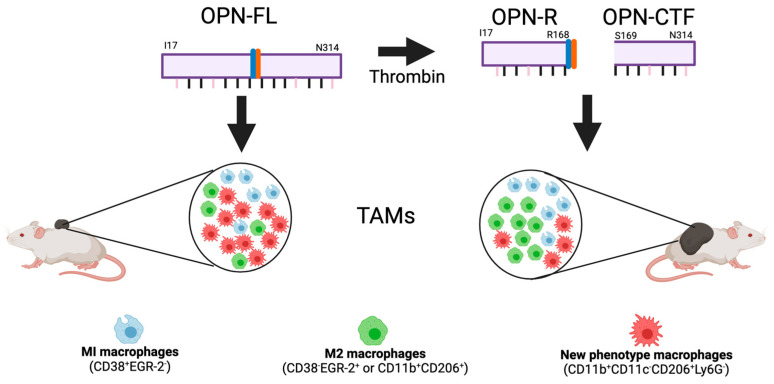
Model of thrombin cleavage fragments of osteopontin (OPN) in promoting cancer. Three subtypes of macrophages were identified amongst tumor-associated macrophages (TAMs) isolated from murine tumors by flow cytometry: M1 (blue, CD38^+^EGR-2^−^), M2 (green, CD38^−^EGR-2^+^ or CD11b^+^CD206^+^), and new phenotype macrophages (red, CD11b^+^CD11c^−^CD206^+^Ly6G^−^). The markers are CD11b: integrin α_M_, CD11c: integrin α_X_, CD38: ADPRC 1, CD206: mannose receptor, EGR-2: early growth response gene-2, and Ly6G: GR-1. Thrombin cleaves full-length OPN (OPN-FL) into OPN-R and OPN-CTF, that down-regulate the host anti-tumor response. The numbers of M1 and M2 subtype macrophages in tumors from mice that were unable to generate OPN-R and OPN-CTF were decreased, while new phenotype macrophages were increased in comparison to wild-type mice able to produce OPN-R. These changes in the phenotype of the TAMs resulted in smaller tumors on mice that do not have OPN or if the OPN gene has a mutation preventing thrombin cleavage. Created with BioRender.com.

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
