# Peer review of "Thrombin Cleavage of Osteopontin and the Host Anti-Tumor Immune Response"

_cancers, 2023, doi:10.3390/cancers15133480_

Round 1

Reviewer 1 Report

The manuscript by Leung et is a review on the role of osteopontin (OPN) and its thrombin-mediated cleavage in tumor growth. The matter is interesting and the review is mostly clearly written. However, the title does not hit the matter of the review and some lacking aspects of OPN activity should be included for further information.

MAJOR POINTS

1) Only a small part of the review is dedicated to the effects of OPN on the anti-tumor immune response. Since most part is dedicated to the effects of OPN on other aspects of tumor biology, I suggest to adjust the title in order to fit the global review content.

2) The review gives only few hints about the OPN activity on the adaptive immune response. I suggest to expand this topic since it may play a crucial role in the anti-tumor immune response.

3) The review gives only few hints about intracellular OPN, whose intriguing activity has been highlighted by the finding that it may play a role in crucial signaling pathways (Nat Immunol. 2015 Jan;16(1):96-106),

4) The review ignores the recent finding that, beside the other several ligands, OPN can bind also ICOSL with effects that are functionally antagonized by ICOS. These interactions may have key effects on tumor cell migration and metastatization, tumor angiogenesis and anti-tumor response (Commun Biol. 2020 Oct 26;3(1):615.).

MINOR POINTS

5) Pg 6 line 187: some text is lacking in the sentence “…to the integrin-binding sites in the sequence...”

6) Pg 10 line 381: the sentence “…where induction HLA-G caused induction of TAMs to produce OPN, which conferred the cancer cells with anti-tumor immunity as well as increasing their proliferation and invasion [108]” is not clear. Please rephrase.

7) Pg.12: Section 9 is not clear and should be reorganized in order to better explain the different results obtained in chemically induced tumors and transplantable tumors.

8) Pg 12 line 469: the authors state that “The lack of OPN expression in the transfected cells increased host anti-tumor macrophage activity” whereas at line 464 they state that “there were more macrophages present in the slow-growing OPN expressing tumors”. These sentences seem to contradict each other.

9) Pg 12 line 473: the sentence “Confirmation of OPN’s role in regulating metastasis came from experiments in which OPN production in C5P2 cells knocked down with ribozymes were more susceptible to the toxic effects of NO produced by inducible NOS in RAW 264.7 cells (a macrophage-like cell line) and, when injected into nude mice, produced fewer metastases” is not clear. Please rephrase

10) Pg 12 Line 476: the sentence “A contradictory effect was observed in OPN-KO mice on increasing chemical tumorigenesis while in other experimental settings, reducing growth and metastasis of implanted tumors” is not clear. Please rephrase.

11) Fig 8 legend: the sentence “The numbers of M1 and M2 subtype macrophages in tumors from mice that were unable to generate OPN-R and OPN-CTF while new phenotype macrophages were increased in comparison to wild type mice able to produce OPN-R” is not clear. Please correct.

12) Pg 20, line 817: the sentence “and the reactions are often use a high enzyme/substrate ratio” is not clear. Please correct.

13) Some acronyms need to by defined, such as CBC (pg 17 line 662), DOAC (pg 19, line 756), NSCLC (pg 19, line 70), PK (pg 20, line 797)

Reviewer 2 Report

Lawerence L Leung et co-authors in the paper submitted to Cancers (titled: Thrombin cleavage of osteopontin regulates host anti-tumor 2 immune response) have evaluated the role of OPN in cancer and the consequences of its cleavage by thrombin.

The review is very interesting.

In order to make reading more fluid, I would recommend shortening the introduction text. Some elements of the introduction, more inherent to the OPN structure, could be moved to paragraph 2. OPN protein structure

The sentence: “to the integrin-binding sites in the sequence 159RGDSVVYGLR168 is carried out by FAM20C, also present in the microsomes [48]. The levels of FAM20C can vary and in the presence of expressed Ras, levels of FAM20C and OPN phosphorylation are reduced [49]” lines 187-189 is not clear

Figure 5 reported different types of cancer (CESC, COAD, LGG, LIHC, LUAD, READ) but in the text, only colon cancer was mentioned and commented.

In relation to OPN expression in hepatocellular carcinoma (pag 8) other articles, pertinent to the topic, could be added:

·       Cabiati M, Gaggini M, Cesare MM, Caselli C, De Simone P, Filipponi F, Basta G, Gastaldelli A, Del Ry S. Osteopontin in hepatocellular carcinoma: A possible biomarker for diagnosis and follow-up. Cytokine. 2017 Nov;99:59-65. doi: 10.1016/j.cyto.2017.07.004. Epub 2017 Jul 12. PMID: 28711012.

·       Cabiati M, Gaggini M, De Simone P, Del Ry S. Data mining of key genes expression in hepatocellular carcinoma: novel potential biomarkers of diagnosis prognosis or progression. Clin Exp Metastasis. 2022 Aug;39(4):589-602. doi: 10.1007/s10585-022-10164-9. Epub 2022 Apr 16. PMID: 35429302; PMCID: PMC9338913.

·       Cabiati M, Di Giorgi N, Salvadori C, Finamore F, Del Turco S, Cecchettini A, Rocchiccioli S, Del Ry S. Transcriptional level evaluation of osteopontin/miRNA-181a axis in hepatocellular carcinoma cell line-secreted extracellular vesicles. Pathol Res Pract. 2022 Aug 23;238:154088. doi: 10.1016/j.prp.2022.154088. Epub ahead of print. PMID: 36084428.

In Paragraph 7 (OPN mRNA expression in tumor cells and tumor associated cell) the evaluation of hepatocellular carcinoma could be also added and the following references could be added and commented:

·       S. Chae, H.O. Jun, E.G. Lee, S.J. Yang, D.C. Lee, J.K. Jung, K.C. Park, Y.I. Yeom, K.W. Kim Osteopontin splice variants differentially modulate the migratory activity of hepatocellular carcinoma cell lines Int. J. Oncol., 35 (2009), pp. 1409-1416 https://doi.org/: 10.3892/ijo_00000458

·       J. Zhang, H. Guo, Z. Mi, C. Gao, S. Bhattacharya, J. Li, P.C. Kuo EF1A1-actin interactions alter mRNA stability to determine differential osteopontin expression in HepG2 and Hep3B cells Exp. Cell. Res., 315 (2009), pp. 304-312, 10.1016/j.yexcr.2008.10.042

·       H. Guo, C.E. Marroquin, P.Y. Wai, P.C. KuoNitric oxide-dependent osteopontin expression induces metastatic behavior in HepG2 cells Dig. Dis. Sci., 50 (2005), pp. 1288-1298, 10.1007/s10620-005-2775-6

·       In my opinion the Paragraph could be integrated in conclusion and shorten.

ok

Reviewer 3 Report

Comprehensive review on osteopontin, its post-transcriptional/translational modification, role in malignancy, cleavage via thrombin and effect on anti-tumor immunity.

Comments: 

Main concerns:

What is the difference between the two subtypes:

CD11b+CD206+ and the new phenotype macrophages  CD11b+CD11c-CD206+Ly6G-). Since both are CD11 +ve and CD206 +ve and the new phenotype is CD11c -ve Ly6G -ve.??

Also was this new subtype published before? If so what is the reference.

The use of the term “TAMs” in the text in different contexts is confusing in some areas, it would be better to clarify the subtype the authors are referring to in each context, since there are several subtypes in literature:

line 688: The use of the term TAMs in line 688 may be confusing, it is better to say…” tumor suppression mediated by M1 or TAM_M1 type”

Line 701: better to say : switch in TAMs profile from M2 to a different activation phenotype

Line 718: better clarify the subtype of TAMs that are decreased. I assume the authors here refer to M1, is it also the CD11b+ and CD11c- CD206+??

Line 721, I assume the TAMs here are referring to M1 subtype again and may be the CD11b+CD11c-CD206+??

Same in line 874 and other locations. Please clarify subtype of TAMs. The idea of TAMs in general, being pro- or anti tumorigenic is an area of debate in literature, and the safest and least confusing way is to report the exact subtype you are referring to in each context. For example There is a systematic and meta analysis study showing TAM CD206+ve correlation with worse prognosis. PMID: 34298638

References,  should include PMID: 36338570 as a reference of the soluble OPN in cancer and PMID: 33776991 on the transcriptional level of OPN in cancer

Minor comments

Figures: Figure 1 and figure 2,  if already present on pubmed, then referencing the website would be enough. Yet if generated by Biorender specially for this manuscript then please clarify and mention in text.

Figure 4 : bars representative of the posttranscriptional modifications can be much smaller, the whole figure can be smaller.

Figure 5: is this previously published data in ref 64 and 65??

Bullet 4, is this physiologic function of osteopontin, if so please mention so, and would be more informative to mention the specific model/cell type in which the function ( binding to extracellular molecules) has been demonstrated.

Bullet 5 should be merged with bullet #6

Bullet 8: should say : OPN in cell culture tumor models

Abstract: The role of OPN in metastases, angiogenesis and metastases is repeated lines 11 and 15. Please take off repetition. 

Reviewer 4 Report

The review manuscript by Leung et al. outlines the role of osteopontin (OPN) in tumor immunity, including that in the tumor microenvironment and therapeutics targeting OPN as tumor immunotherapy. Overall, the general information of OPN (e.g., structure, splicing variants, post-transcriptional modifications) are extensively described. However, this part is oversized and related to neither thrombin cleavage nor tumor biology, considering the title of the review, “Thrombin cleavage of OPN regulates host anti-tumor immune response”. Besides, the former part of the review has been covered more thoroughly in other recent reviews (e.g., Moorman et al., Cancers 2020). Moreover, the specific role of cleaved OPN in tumor immunity was not summarized, rather difficult to interpret the contents without reading the original paper. Together, this review does not match the title and the content, which is insufficient to attract readers' interest.

Specific comments

1.     The detailed review of structure and post-translational modifications of OPN in sections 2 and 3 provides many details that are not relevant to physiological function and tumor evasion of host immunity.

2.     The text in section 4 focuses on the OPN function but does not address the consequence of binding OPN with integrins.

3.     In section 7, the authors focused on OPN mRNA expression in tumor cells and tumor-associated cells. More work needs to be done reviewing the latest progress regarding SPP1+ macrophages in the tumor microenvironment.

Yang, Q. et al. Single-Cell RNA Sequencing Reveals the Heterogeneity of Tumor-Associated Macrophage in Non-Small Cell Lung Cancer and Differences Between Sexes. Front Immunol 12, 756722 (2021).

Leader, A. M. et al. Single-cell analysis of human non-small cell lung cancer lesions refines tumor classification and patient stratification. Cancer Cell 39, 1594-1609.e12 (2021).

Dong, B., Wu, C., Huang, L. & Qi, Y. Macrophage-Related SPP1 as a Potential Biomarker for Early Lymph Node Metastasis in Lung Adenocarcinoma. Front Cell Dev Biol 9, 739358 (2021).

4.     In sections 11 and 12; Thrombin cleavage of OPN has been shown to have a key role in several models such as cardiac fibrosis, HSC niche, microglia, and liver fibrogenesis. But few reports regarding tumor biology so far. The only evidence is that the concentration of cleaved OPN was elevated in GBM patients. However, how cleaved OPN plays a role in tumor growth, induction of tumor immunity, and tumor outcome remain unclear. In addition, the authors recently generated OPN-KI mice and showed a similar phenotype to OPN-KO mice, suggesting that OPN cleavage is necessary for the progression of tumors. But these are self-citation and poor discussion in comparison with other papers.

5.     In section 13, the authors described the role of anti-coagulants or anti-OPN antibodies in inhibiting OPN cleavage. Is there any evidence of their usage in tumor models?

The text is also very difficult to read, in part due to fragmental descriptions of each topic. Many sentences must be read multiple times to discern the intended point.

Round 2

Reviewer 1 Report

 In the revised version of the manuscript the authors fulfilled most of the suggestions of this referee.

However, few minor points can be raised:

MINOR POINTS

1) Pg3 line 83: urolithiasis cannot be cited as a typical example of autoimmune disease. OPN is involved in many typical autoimmune diseases.

2) Pg 8 line 253: the sentence “OPN acts a functional antagonist of ICOS binding to ICOSL” is misleading since OPN does not antagonize ICOS binding to ICOSL. By contrast, OPN binding to ICOSL promotes cell migration and tumor angiogenesis, but ICOS triggering of ICOSL dominantly inhibit these effects.

3) Pg 11 line 442. The sentence “OPN and MMP9 were no different” should be corrected with “OPN and MMP9 were not different”    

4) Pg 13 line 527 The sentence “finding that a several cell types” should be corrected with “finding that several cell types”

5) Pg 13 line 549: the sentence “These TAMS then produced OPN, which conferred the cancer cells with anti-tumor immunity “is still not clear. Please rephrase.

6) Pg 26 line 1112: the sentence “a Cochrane systematic review that there is a survival benefit of parenteral heparin” should be corrected with “a Cochrane systematic review showed that there is a survival benefit of parenteral heparin”

Author Response

Please find our responses in Italics in the attached document.

Reviewer 4 Report

In the revised manuscript, the authors responded to each concern raised by this reviewer. By contrast, the word length of the revised manuscript has become longer than the initially submitted one, which can be difficult for readers in the field of oncology and cancer immunology to deeply understand the contents. Thus, this reviewer recommends that some sections, particularly not directly related to cancer immunology, need to be summarized to be understandable in the review. 

Comment to response 1

We are unaware of a study that definitely proves that a particular interaction between OPN and one of its ligands is key for its unknown physiological function or for tumor evasion of host anti-tumor immunity. Similarly, the effects of OPN on the outcome of any specific cell assay have not been shown to be essential for its unknown physiological function or for tumor evasion of host anti-tumor immunity. The details of OPN structure and its post-translational modifications are given in our review because the modifications vary between cell types and those differences modulate functions some of which are involved with tumor evasion of host anti-tumor immunity. 

The reviewer understood that the structure and post-translational modifications of OPN are closely related to OPN cleavage by thrombin. However, given the length of the entire text, only the evidence related to cancer and cancer immunity should be summarized in these sections.

Comment to response 2

Three paragraphs describing the signaling pathways initiated by OPN binding to its receptors have been added to the end of Section 4.

Again, these consequences are better to be summarized.

Comment to response 4

In the models noted by the reviewer, the evidence for that the thrombin cleavage fragments are mechanistically critical for the development of the phenotype is lacking. In cardiac fibrosis, Herum et al (2020) show the presence of the thrombin cleaved fragments of OPN in the aortic banding model of cardiac fibrosis by Western. They show that OPN-R induces a 1.5x-fold increase in expression of three collagen genes in neonatal cardiac fibroblasts while a peptide representing the C-terminal sequence of OPN-R led to a 3x-fold increase in colagenA1 gene expression in adult cardiac fibroblasts. They show that syndecan 4 binds OPN but and thereby protects it from thrombin cleavage. Thus, the data only shows a correlation between generation of OPN cleavage fragments and fibrosis.

In the HSC model, Grassinger et al (2009) reported that upon GM-CSF challenge OPN-KO mice mobilized more WBCs including progenitor cells than WT mice. The data shows that the thrombin cleavage fragments can affect HSC properties but only correlate with the effects. In contrast to expectations, in our OPN-KI mice without GM-CSF challenge, Peraramelli et al (2022 supplementary material) showed that there was no difference in CBC between OPN-KI, OPN-KO and WT mice at any age or either gender. There was no increase in progenitors or in WBC count on GM-CSF challenge (unpublished data). 

Similar arguments can be made about the other models that the reviewer mentions: microglia and liver. In each case there is evidence presented that thrombin cleavage of OPN is occurring but, to our reading of the papers, there is no proof that the thrombin cleavage of OPN is integral to their mechanism.

The authors should stick to the role of thrombin cleavage of OPN in tumor immunity. If the role of the cleavage is not relevant even in non-cancer models, the authors should omit them.

Comments on the Quality of English Language

We have edited the manuscript to improve readability.

It contains many deviations from the title and abstract, and the description is not intended for readers in the field of oncology and tumor immunology, which resulted in difficulty in reading the manuscriptAgain, this review should be described only as related to the role of thrombin cleavage of OPN in tumor immunity, as described in the title and abstract.

Author Response

Please find our responses in the attached document.
